# Manipulating a Learning Defender and Ways to Counteract

**Jiarui Gan**
University of Oxford
Oxford, UK
jiarui.gan@cs.ox.ac.uk

**Qingyu Guo**
Nanyang Technological University
Singapore
qguo005@e.ntu.edu.sg

**Long Tran-Thanh**
University of Southampton
Southampton, UK
l.tran-thanh@soton.ac.uk

**Bo An**
Nanyang Technological University
Singapore
boan@ntu.edu.sg

**Michael Wooldridge**
University of Oxford
Oxford, UK
mjw@cs.ox.ac.uk

## Abstract

In Stackelberg security games when information about the attacker's payoffs is uncertain, algorithms have been proposed to *learn* the optimal defender commitment by interacting with the attacker and observing their best responses. In this paper, we show that, however, these algorithms can be easily manipulated if the attacker responds untruthfully. As a key finding, attacker manipulation normally leads to the defender learning a *maximin strategy*, which effectively renders the learning attempt meaningless as to compute a maximin strategy requires no additional information about the other player at all. We then apply a game-theoretic framework at a higher level to counteract such manipulation, in which the defender commits to a *policy* that specifies her strategy commitment according to the learned information. We provide a polynomial-time algorithm to compute the optimal such policy, and in addition, a heuristic approach that applies even when the attacker's payoff space is infinite or completely unknown. Empirical evaluation shows that our approaches can improve the defender's utility significantly as compared to the situation when attacker manipulation is ignored.

## 1   Introduction

Stackelberg security games (SSGs) are Stackelberg game models developed for deriving optimal security resource allocation in strategic scenarios. In the AI community, a line of work applying SSG models forms the algorithmic basis of resource scheduling systems, that are in use by the Los Angeles Airport, the US Cost Guard, the Federal Air Marshal Service, etc, to assist in protecting high-profile infrastructures, and public and natural resources [21].

The standard solution concept of SSG, the *strong Stackelberg equilibrium* (SSE) captures the situation where a defender (the leader) commits to her optimal strategy, assuming that an attacker (the follower) will respond optimally to her commitment. There are many algorithms designed to compute SSEs in different SSG models when complete information about the attacker's type (i.e., his payoff parameters) is provided. While payoff information may be incomplete in many real environments, algorithms were also designed for the defender to *learn* the optimal commitment through interacting with the attacker: by committing to a series of carefully chosen defender strategies and observing the attacker's best responses to these strategies [11, 4, 8, 19, 17]. The optimality of the learned commitment thus relies crucially on the assumption of a truthful attacker, one who responds to the defender's commitment optimally according to their actual payoffs. Unfortunately, when there is no guarantee that the

attacker will indeed be truthful, a strategic attacker can easily manipulate the learning algorithm by using fake best responses — typically by imitating the responses of a different attacker type. The defender will then learn a commitment that is optimal with respect to the imitated type but, very likely, suboptimal with respect to the true attacker type. As we will show in this paper, the attacker is often incentivized to imitate a type that makes the game zero-sum; a credulous defender would then only learn a *maximin strategy* (i.e., the optimal commitment in a zero-sum game). Effectively, the learning attempt now becomes meaningless: to compute a maximin strategy, one needs no additional information about the other player's payoffs at all!

Driven by this issue, we study what can be done to reduce the defender's loss due to attacker manipulation. We apply a game-theoretic framework at a higher level. In the framework, the defender commits to a *policy* that specifies her strategy commitment according to the learned information. A strategic attacker then takes into account the defender's policy, choosing optimally what he wants the defender to learn so that the policy outputs a strategy that benefits him the most. We make several other contributions under this framework. (i) We propose a novel quality measure of the defender's policy and argue why it is a reasonable choice in the context of SSGs. (ii) We develop a polynomial-time algorithm to compute the optimal policy with respect to this quality measure, as well as (iii) a heuristic approach which applies even when the attacker type space is infinite or completely unknown. The heuristic approach is inspired by the famous *quantal response* model that was initially proposed to model bounded rationality of human players. It suggests a rationality of behaving in a "boundedly rational" manner in the presence of attacker manipulation. (iv) Finally, through empirical evaluation we show that our approaches can improve the defender's utility significantly in randomly generated games, as compared to the situation when attacker manipulations are ignored.

Our work follows some recent research effort on understanding manipulation of leader learning algorithms in general Stackelberg games [7] and shed light on the same issue in SSGs. The SSG model offers us an appropriate level of specification that enables us to derive a richer set of results than from a general model (consider, e.g., when the interests of the leader and the follower completely align, there is simply no incentive for the follower to manipulate the leader), while it also captures sufficiently many real-world applications of significant practical value.

**Additional Related Work**   Apart from [7], manipulation of leader learning algorithms remains largely an under-explored topic, though there are many papers focusing on the design learning algorithms for the leader. In addition to the aforementioned efforts to learn the optimal leader commitment against a fixed but unknown follower type [11, 4, 8, 19, 17], a couple of papers also take the regret-minimization perspective and design online learning algorithms for the leader to use in the adversarial setting [1, 22]. Our work can be seen as a middle-ground approach between the overoptimistic assumption of a truthful follower adopted by the former line of work and the pessimistic assumption of a worst-case opponent by the latter. Our approach to deal with attacker manipulation is, in a nutshell, to reduce information uncertainty by acquiring additional information while bearing in mind that information acquired may be manipulated by the attacker. There are also approaches to deal with uncertainties without additional information retrieval attempts; they are immune to manipulation as a result. In particular, algorithms were designed to compute robust leader strategies when the leader can bound the follower's payoffs in certain intervals [11, 10, 14], or knows a probability distribution of the follower's type [6, 16, 9, 18]. Follower manipulation is not a concern in applying these approaches but leader strategies yielded are weaker. In a broader sense, our work is also related to poisoning attacks in adversarial machine learning, where an attacker manipulates the training data (in our model, their payoffs) to undermine the performance of learning algorithms; see, e.g., pioneering work in this area [2, 3] and some recent surveys [5, 12].

## 2   SSG Preliminaries

An SSG is played between a *defender* (the leader) and an *attacker* (the follower). The defender allocates $m$ security resources to a set of targets $T = \{1, \ldots, n\}$ (without loss of generality, $n > m$), and the attacker chooses a target to attack. In the pure strategy setting, an attack on a target $i$ is unsuccessful as long as one resource is allocated to $i$, in which case the attacker receives a penalty $p_i^{\mathrm{a}}$ and the defender a reward $r_i^{\mathrm{d}}$. Otherwise, i.e., when no resource is allocated to $i$, the attack is successful, in which case the attacker receives a reward $r_i^{\mathrm{a}}$ and the defender a penalty $p_i^{\mathrm{d}}$. We say that a target is *protected*, or *covered*, if at least one resource is assigned to it; and *unprotected*, or

*uncovered*, otherwise. It is assumed that $r_i^{\mathrm{a}} > p_i^{\mathrm{a}}$ and $r_i^{\mathrm{d}} > p_i^{\mathrm{d}}$ for all $i$, so the attacker always prefers a successful attack and the defender prefers the opposite.

The defender can further randomize the resource allocation and commit to a mixed strategy, i.e., a probability distribution over pure strategies. The structure of SSGs allows a defender mixed strategy to be represented more compactly as a *coverage (vector)* $\mathbf{c} = (c_i)_{i \in T}$, with each $c_i$ representing the probability that target $i$ is protected. We will stick to this representation and use the terms *coverage* and *defender mixed strategy* interchangeably. Under the constraint that the defender can use at most $m$ resources, the set of feasible mixed strategies available for the defender to use is $\mathcal{C} = \{\mathbf{c} \in \mathbb{R}^n : 0 \le \mathbf{c} \le 1, \sum_{i \in T} c_i \le m\}$; provably, any coverage vector in $\mathcal{C}$ can be implemented as a distribution of pure strategies each involving at most $m$ resources, and any such distribution results in a coverage vector in $\mathcal{C}$. When the defender plays a mixed strategy $\mathbf{c}$ and the attacker attacks target $i$, let $u^{\mathrm{d}}(\mathbf{c}, i)$ and $u^{\mathrm{a}}(\mathbf{c}, i)$ be the expected utilities of the defender and the attacker, respectively. With slight abuse of notation, we write

$$u^{\mathrm{d}}(\mathbf{c}, i) = u^{\mathrm{d}}(c_i, i) = c_i \cdot r_i^{\mathrm{d}} + (1 - c_i) \cdot p_i^{\mathrm{d}}, \tag{1}$$

$$u^{\mathrm{a}}(\mathbf{c}, i) = u^{\mathrm{a}}(c_i, i) = (1 - c_i) \cdot r_i^{\mathrm{a}} + c_i \cdot p_i^{\mathrm{a}}. \tag{2}$$

It is worth noting that $u^{\mathrm{d}}(\mathbf{c}, i)$ is strictly increasing with respect to $c_i$, and $u^{\mathrm{a}}(\mathbf{c}, i)$ strictly decreasing. By the standard assumption, in an SSG the attacker is able to observe the defender's mixed strategy through surveillance before he launches an attack, but the instantiated pure strategy is not observable.

The *strong Stackelberg equilibrium* (SSE) is the standard solution concept of SSGs. In an SSE, the defender commits to an optimal mixed strategy, taking into account that the attacker will observe this strategy and respond optimally. It is assumed that ties are broken in favor of the defender when the attacker has multiple best responses; hence, without loss of generality we can assume that the attacker always responds by playing a *pure* strategy. The assumption is justified by the fact that the attacker's strict preference to the favored target can be induced if the defender reduces the coverage of this target by an infinitesimal amount.[1] Formally, a strategy profile $(\hat{\mathbf{c}}, \hat{i})$ forms an SSE if:

$$(\hat{\mathbf{c}}, \hat{i}) \in \arg\max_{\mathbf{c} \in \mathcal{C}, i \in \mathrm{BR}(\mathbf{c})} u^{\mathrm{d}}(\mathbf{c}, i),$$

where $\mathrm{BR}(\mathbf{c}) = \arg\max_{i \in T} u^{\mathrm{a}}(\mathbf{c}, i)$ denotes the set of attacker best responses to $\mathbf{c}$. An SSE always exists and can be computed in polynomial time, e.g., using a multiple-LP approach [6].

We will refer to a full set of attacker payoffs $(\mathbf{r}^{\mathrm{a}}, \mathbf{p}^{\mathrm{a}})$ as an *attacker type*. To distinguish, we extend (2) and define $u_\theta^{\mathrm{a}}(\mathbf{c}, i) = (1 - c_i) \cdot x_i + c_i \cdot y_i$ to be the utility function parameterized by attacker type $\theta = (\mathbf{x}, \mathbf{y})$. Definition of the best response set is extended likewise, with $\mathrm{BR}_\theta(\mathbf{c}) = \arg\max_{i \in T} u_\theta^{\mathrm{a}}(\mathbf{c}, i)$. We will refer to an SSE in a game where the attacker's type is $\theta$ an *SSE on attacker type* $\theta$.

**Example 1.** Consider an SSG where the defender allocates one security guard to protect two targets $A$ and $B$. The defender has three pure strategies: to assign the guard to protect $A$ or $B$, or to send the guard on vacation; the corresponding mixed strategy space is $\mathcal{C} = \{\mathbf{c} \in \mathbb{R}^2 : 0 \le \mathbf{c} \le 1, c_1 + c_2 \le 1\}$. The attacker can choose to attack $A$ or $B$. In this game, the targets are of equal importance to the defender: a successful attack on any target $i \in \{A, B\}$ results in utility $p_i^{\mathrm{d}} = -1$ for the defender, and an unsuccessful one results in $r_i^{\mathrm{d}} = 0$. For the attacker, the payoffs are $r_A^{\mathrm{a}} = 3$, $r_B^{\mathrm{a}} = 1$, and $p_A^{\mathrm{a}} = p_B^{\mathrm{a}} = 0$. The bi-matrix representation of the game is shown below, in which the defender and the attacker are the row and column players, respectively.

|  | *attack A* | *attack B* |
|---|---|---|
| *protect A* | 0,  0 | -1,  1 |
| *protect B* | -1,  3 | 0,  0 |
| *protect ∅* | -1,  3 | -1,  1 |

The SSE of this game can be identified using the indifference rule, i.e., by identifying a point where the attacker is indifferent of attacking $A$ and $B$, while the defender cannot improve coverage of the targets any further (however, not in every game can an SSE be found in this way). In the only SSE of this game, the defender protects $(A, B)$ with probabilities $\mathbf{c} = (\frac{3}{4}, \frac{1}{4})$ (which is equivalent to a mixed strategy $\mathbf{x} = (\frac{3}{4}, \frac{1}{4}, 0)$ as in the bi-matrix representation), to which the attacker finds his best responses to be $\mathrm{BR}(\mathbf{c}) = \{A, B\}$ and, by the SSE assumption, breaks the tie in favor of the defender by attacking $A$. The defender gets utility $u^{\mathrm{d}}(\mathbf{c}, A) = -\frac{1}{4}$ and the attacker gets $u^{\mathrm{a}}(\mathbf{c}, A) = \frac{3}{4}$.

# 3 Manipulating a Learning Defender

We investigate how attacker manipulation can take place. Let us begin with a warm-up example.

**Example 2.** Consider now the attacker in Example 1 pretends to have payoff $r_A^a = 1$ (all other parameters remain the same) and "best" responds to queries of the defender's learning algorithm according to this fake parameter. Let the resultant fake attacker type be $\beta$. The defender will be misled into learning an SSE on type $\beta$, in which her commitment is $\tilde{\mathbf{c}} = (\frac{1}{2}, \frac{1}{2})$. We have $\mathrm{BR}_\beta(\tilde{\mathbf{c}}) = \{A, B\}$. The attacker can respond (still with ties broken in favor of the defender) by attacking $A$, and this results in the attacker's utility to increase to $\frac{3}{2}$, but the defender's to drop to $-\frac{1}{2}$. There is a loss of $\frac{1}{4}$ for the defender compared to the truthful situation! Note that the attacker behaves consistently according to the fake type $\beta$ even after having misled the defender into learning the fake commitment. Hence, there is no way to distinguish him from a truthful type-$\beta$ attacker.

In the above example, the attacker actually lies to the defender that the game they are playing is zero-sum. It turns out that this is not a coincidence specific to this example but a general phenomenon in SSGs. We show next that it is always optimal for the attacker to mislead the defender into playing her *maximin strategy*. A maximin strategy $\overline{\mathbf{c}}$ maximizes the defender's utility against the worst possible attacker type, i.e., $\overline{\mathbf{c}} \in \arg\max_{\mathbf{c}} \min_i u^d(\mathbf{c}, i)$; it is exactly the defender's optimal commitment in a zero-sum game (see Lemma 13 in the appendix).

A couple of "disclaimers" would be appropriate before we delve into our analysis: First, in line with previous work (e.g., [11, 4, 17]), we only consider the players' utilities in the (fake) SSE the defender learns. The cost incurred for both players during the learning process is omitted as we expect the learning algorithm to run efficiently and the learned SSE to repeat in sufficiently many rounds. Without loss of generality, we view the learning process as a reporting step in which the attacker simply *reports* his type to the defender. To manipulate the defender, the attacker reports a fake type, and we refer to this as his *reporting strategy*.

Second, we assume that the attacker behaves consistently according to the reported type. This means that the attacker may be playing a fake best response — hence, a suboptimal one — in the learned SSE and may thus exploit the defender for an even higher utility by switching back to his true best response. Nevertheless, since such a change in his behavior will inevitably make the defender aware of the manipulation and further complicates the interaction, we ignore the possibility of such a behavior change and adopt this cleaner model to capture the essence of the manipulation problem.

**Optimal Attacker Report** The following program computes the optimal reporting strategy of a type-$\theta$ attacker.

$$\max_{\beta, \mathbf{z}, t} \quad u_\theta^a(\mathbf{z}, t) \tag{3}$$

$$\text{s.t.} \quad (\mathbf{z}, t) \in \arg\max_{\mathbf{c} \in \mathcal{C}, \, i \in \mathrm{BR}_\beta(\mathbf{c})} u^d(\mathbf{c}, i) \tag{3a}$$

$$\beta \in \Theta \tag{3b}$$

Here $\Theta = \{(\mathbf{r}^a, \mathbf{p}^a) \in \mathbb{R}^{n \times n} : r_i^a > p_i^a \text{ for all } i \in T\}$ contains all types that adhere to the basic assumption that an attacker always prefers a successful attack (we will show a stronger result that allows for other specifications of $\Theta$). In the program, the attacker reports a fake type $\beta$ that results in the defender to learn an SSE $(\mathbf{z}, t)$ on type $\beta$ (by (3a)). An optimal solution thus yields a reporting strategy for a type-$\theta$ attacker, that maximizes his *true* utility as the objective function specifies.

Using Program (3), we show with Theorem 3 that it is always optimal for the attacker to mislead the defender into playing her maximin strategy. This result is surprising as the defender essentially learns nothing: she can well compute the maximin strategy without any additional knowledge about the attacker's payoffs. We present a proof sketch. All omitted proofs can be found in the appendix.

**Theorem 3.** *There exists an optimal solution* $(\beta, \mathbf{z}, t)$ *of Program* (3) *such that* $\mathbf{z}$ *is a maximin strategy of the defender, i.e.,* $\mathbf{z} \in \arg\max_{\mathbf{c} \in \mathcal{C}} \min_{i \in T} u^d(\mathbf{c}, i)$.

*Proof sketch.* Let $\overline{\mathbf{c}}$ be a maximin strategy of the defender and $\overline{u}$ be her maximin utility. Consider a solution $(\beta, \mathbf{z}, t)$ such that: $z_i = \max\left\{0, \frac{\overline{u} - p_i^d}{r_i^d - p_i^d}\right\}$ for all $i \in T$; $t \in \mathrm{BR}_\theta(\mathbf{z})$; and $\beta = (\mathbf{r}, \mathbf{p})$, such that $r_i = \begin{cases} -p_i^d, & \text{if } i \neq t \\ -\min\{p_t^d, \overline{u}\}, & \text{if } i = t \end{cases}$, and $p_i = -r_i^d$ for all $i \in T$. It can be verified that $\mathbf{z}$ is indeed also a maximin strategy of the defender and $(\beta, \mathbf{z}, t)$ is an optimal solution. $\qquad \square$

Theorem 3 can be further strengthened under the assumption that the defender's maximin strategy $\overline{\mathbf{c}}$ is fully mixed, i.e., $0 < \overline{c}_i < 1$ for all $i$. (In this case the maximin strategy is also unique; see Lemma 15 in the appendix.) The assumption is mild as it is normally expected that no target would be too worthless to the extent that the defender would leave it wide open for the attacker to attack, while on the other hand resources are normally insufficient to allow any target to be fully protected. The strengthening is two-fold: (i) under the additional assumption, the defender's maximin strategy is her *only* SSE strategy induced by any optimal attacker report, so the equilibrium selection issue that arises when a reported type induces multiple SSEs is avoided; (ii) one optimal attacker report, in particular, is the type that makes the game zero-sum, so our result holds even for a more stringent specification of $\Theta$ (e.g., when the defender has more precise knowledge about possible attacker types) as long as $\Theta$ contains the zero-sum attacker type. (Indeed, it is very natural for an attacker to have the zero-sum type given the adversarial nature of SSGs.) We state the result in Theorem 4.

**Theorem 4.** *Suppose $\overline{\mathbf{c}}$ is a maximin defender strategy and it is fully mixed, i.e., $0 < \overline{c}_i < 1$ for all $i \in T$. Let $(\beta, \mathbf{z}, t)$ be an arbitrary optimal solution of Program* (3). *For every SSE $(\hat{\mathbf{c}}, \hat{i})$ on type $\beta$, it holds that $\hat{\mathbf{c}} = \overline{\mathbf{c}}$. In addition, there exists an optimal solution $(\beta', \mathbf{z}', t')$ such that $\beta' = (-\mathbf{p}^{\mathrm{d}}, -\mathbf{r}^{\mathrm{d}})$.*

## 4  Handling Attacker Manipulation — A New Playbook

Recall our analysis. The key to the success of the attacker's trick is the naive playbook the defender follows — to always play the learned optimal commitment *as is*. It appears that the defender can be more strategic. Consider Example 2. Suppose the defender tweaks her strategy slightly, playing $\tilde{\mathbf{c}} = (\frac{1}{2}, \frac{49}{100})$ even when she learns that $(\frac{1}{2}, \frac{1}{2})$ is optimal. The attacker, who imitates type $\beta$ (that makes the game zero-sum), should then attack $B$ as now the best response set of $\beta$ becomes $\mathrm{BR}_\beta(\tilde{\mathbf{c}}) = \{B\}$. The attacker obtains utility $\frac{1}{2}$, which is even lower than his utility $\frac{3}{4}$ in the truthful situation. Therefore, if the defender commits to playing, e.g., $(c_1, c_2 - \frac{1}{100})$ whenever she learns that $(c_1, c_2)$ is optimal, the attacker will at least lose the incentive to mislead the defender into playing her maximin strategy. The question then becomes: what is the best the defender can achieve by revising her playbook in similar ways? We formalize this problem as finding an optimal *policy* to commit to.

**Committing to a Policy**  Formally, a policy is a function $\pi : \Theta \to \mathcal{C} \times T$ that maps a reported attacker type to an *outcome* $(\mathbf{c}, i) \in \mathcal{C} \times T$. An outcome $(\mathbf{c}, i)$ is a strategy profile consisting of a defender strategy $\mathbf{c}$ and a best response $i \in \mathrm{BR}_\theta(\mathbf{c})$ of the reported attacker type $\theta$.[2] As an example, the way the defender plays when she ignores attacker manipulation can itself be viewed as a policy that maps every reported type $\theta$ to an SSE on $\theta$; we will refer to this policy as the *SSE policy*.

We assume that a policy can be observed or learned by the attacker through constant interaction with the defender, or the defender can simply announce it to the attacker. In response, a strategic attacker of true type $\theta$ chooses to report an optimal type $\beta^* \in \arg\max_{\beta \in \Theta} u^{\mathrm{a}}_\theta(\pi(\beta))$ that will maximize his utility in the outcome of the policy. At a higher level, this can be seen as a Stackelberg game in which the defender commits to a policy and the attacker reports optimally in response to this commitment.

To find the optimal policy, we need a good measure of the quality of a policy. When there is no other prior information about the attacker's type, the worst-case analysis seems to be appropriate and a straightforward choice of quality measure is the utility the defender obtains when playing against the worst attacker type. However, as Proposition 5 suggests, this measure disallows us to well distinguish the quality of many policies. Specifically, when playing against the zero-sum attacker type, no feasible policy can achieve anything better than the maximin utility, so if we take the worst-case utility as the measure, the quality of all policies would be hindered by this attacker type, and the SSE policy, which achieves exactly the maximin utility in the worst case, would then be the best policy we can hope for. Essentially, there will be no room for improvement other than letting the attacker lie.

**Proposition 5.** *Let $\overline{\mathbf{c}}$ be a maximin strategy of the defender, and $\overline{u} = \min_{i \in T} u^{\mathrm{d}}(\overline{\mathbf{c}}, i)$ be the maximin utility. For any policy $\pi$, let $\gamma \in \arg\max_{\theta \in \Theta} u^{\mathrm{a}}_\beta(\pi(\theta))$ be an optimal report of a type-$\beta$ attacker, $\beta = (-\mathbf{p}^{\mathrm{d}}, -\mathbf{r}^{\mathrm{d}})$. Then $u^{\mathrm{d}}(\pi(\gamma)) \leq \overline{u}$.*

This is unreasonable: even in the truthful situation, it is impossible for the defender to achieve more than the maximin utility when the attacker is of the zero-sum type, so it would be unfair to underrate

a policy simply because it underperforms against the zero-sum type. For this reason, we propose an alternative measure, termed *the efficiency of a policy (EoP)*, which takes into consideration the hardness of playing against each attacker type in the truthful setting. As in Definition 6, the EoP is the worst-case ratio between the utility the defender obtains and what she should have obtained had the attacker been truthful. A higher EoP indicates a smaller loss due to attacker manipulation, and the value of the EoP always lies between $0$ and $1$ according to Proposition 7. For the EoP to be meaningful, we shift all payoffs to be non-negative. Without loss of generality, we will hereafter also assume $\Theta$ — previously defined as the set of types the attacker is allowed to report — to be also the set of possible attacker types, which is common knowledge to both players.

**Definition 6** (**EoP**). For each $\theta \in \Theta$, let $\beta_\theta^\pi = \arg\max_{\beta \in \Theta} u_\theta^a(\pi(\beta))$ be the attacker's optimal reporting strategy in response to a policy $\pi$ (tie-breaking in favor of the defender). The efficiency of $\pi$ *on attacker type* $\theta$ is $\mathrm{EoP}_\theta(\pi) = \frac{u^d(\pi(\beta_\theta^\pi))}{\hat{u}(\theta)}$, where $\hat{u}(\theta) = \max_{\mathbf{c} \in \mathcal{C}, i \in \mathrm{BR}_\theta(\mathbf{c})} u^d(\mathbf{c}, i)$ is the defender's utility in an SSE on type $\theta$. The (overall) efficiency of $\pi$ is $\mathrm{EoP}(\pi) = \min_{\theta \in \Theta} \mathrm{EoP}_\theta(\pi)$.

**Proposition 7.** $\mathrm{EoP}(\pi) \in [0, 1]$ *for any feasible defender policy* $\pi$.

Another challenge we face is the representation of a policy, which is a function to be optimized. We follow a modeling approach in the literature and consider a discrete version of the problem where the set $\Theta$ of attacker types is finite. This approach has been widely adopted to model Bayesian games (e.g., [6, 16, 9, 18]). A finite type set can be seen as an approximation to the continuous type space, while in some scenarios attacker types might also be discrete by nature. For example, in defense against poaching, payoffs of the poachers may depend on the type of animal products they are interested in, which falls in a finite set. In addition to this approach, we also propose a heuristic policy that applies to an infinite or even an unknown type set. We present these approaches next.

# 5 Computing the Optimal Policy

## 5.1 Optimal Policy for Finite Attacker Types

When $\Theta$ is a finite set, a defender policy can be represented as a list of $\lambda = |\Theta|$ outcomes; we will therefore also write a policy as $\pi = (\mathbf{c}^\theta, i^\theta)_{\theta \in \Theta}$, meaning that $\pi(\theta) = (\mathbf{c}^\theta, i^\theta)$ for each $\theta \in \Theta$. Our analysis reveals that to compute the EoP maximizing policy is NP-hard in general Stackelberg games (see Section D in the appendix), but thanks to the special utility structure of SSGs, the problem admits a polynomial-time algorithm when the underlying game is an SSG.

We consider the decision version of the optimization problem: for a given value $\xi$, decide whether any defender policy $\pi$ achieves $\mathrm{EoP}(\pi) \geq \xi$. Trivially, once we have an efficient algorithm for this decision problem, the best EoP can be found efficiently using binary search (in particular, we already know that the value always lies in $[0, 1]$). Our algorithm for this decision problem, presented as Algorithm 1, is constructive and produces a satisfying policy when there exists one. In the remainder of this section, we will let $\Theta = \{\theta_1, \ldots, \theta_\lambda\}$ such that $\theta_1, \ldots, \theta_\lambda$ are ordered by the utility they offer the defender in an SSE, i.e., $\hat{u}(\theta_1) \geq \hat{u}(\theta_2) \cdots \geq \hat{u}(\theta_\lambda)$; the order can be obtained efficiently given that an SSE can be computed in polynomial time. We call a policy $\ell$-*compatible* if truthful report is incentivized for every attacker type $\theta_j$, $j \leq \ell$ (Definition 8).

The correctness of Algorithm 1 is shown via Theorem 10. Briefly speaking, Algorithm 1 can be viewed as a process of repeatedly replacing the $\ell$-th outcome of a satisfying policy (suppose we are given one) with the outcome generated in the $\ell$-th iteration of Step 3. The observation in Lemma 9 ensures that the new policy obtained after every replacement will still be a satisfying one. Hence, eventually, we will obtain a satisfying policy that consists of outcomes generated all through the algorithm, and this means that we do not actually need to be provided a satisfying policy to begin with. Interestingly, the policy generated by Algorithm 1 is also *incentive compatible (IC)* ($\lambda$-compatible as in Definition 8); it always incentivizes the attacker to report their true type.

**Definition 8.** A policy $\pi$ is $\ell$-*compatible* ($0 \leq \ell \leq \lambda$), if in response to $\pi$, it is optimal for every attacker type $\theta \in \{\theta_1, \ldots, \theta_\ell\}$ to report truthfully, i.e., $u_\theta^a(\pi(\theta)) \geq u_\theta^a(\pi(\beta))$ for all $\beta \in \Theta$.

**Lemma 9.** *Let $\pi$ be the policy generated in Step 3 of Algorithm 1. Suppose that there exists an $(\ell-1)$-compatible policy $\pi^*$, $\mathrm{EoP}(\pi^*) \geq \xi$. Then the policy $\tilde{\pi}$, such that $\tilde{\pi}(\theta) = \begin{cases} \pi^*(\theta), & \text{if } \theta \in \Theta \setminus \{\theta_\ell\} \\ \pi(\theta), & \text{if } \theta = \theta_\ell \end{cases}$, is feasible and $\ell$-compatible, and $\mathrm{EoP}(\tilde{\pi}) \geq \xi$.*

---
**Algorithm 1:** Decide if there exists a policy $\pi$ such that $\mathrm{EoP}(\pi) \geq \xi$.

---

1. For each $\theta \in \Theta$, compute an SSE $(\hat{\mathbf{c}}^\theta, \hat{i}^\theta)$ on type $\theta$. Let $\hat{u}(\theta) = u^{\mathrm{d}}(\hat{\mathbf{c}}^\theta, \hat{i}^\theta)$.

2. Sort attacker types in $\Theta$ by $\hat{u}(\theta)$, so that $\hat{u}(\theta_1) \geq \hat{u}(\theta_2) \cdots \geq \hat{u}(\theta_\lambda)$, $\lambda = |\Theta|$.

3. For each $\ell = 1, \ldots, \lambda$, let $\pi(\theta_\ell) = (\mathbf{z}, t)$, where $z_i = \min\{\hat{c}_i^{\theta_\ell}, h_i\}$, $t = \mathrm{BR}_{\theta_\ell}(\mathbf{h})$, and
$h_i = \max\left\{ 0, \ \frac{\xi \cdot \hat{u}(\theta_\ell) - p_i^{\mathrm{d}}}{r_i^{\mathrm{d}} - p_i^{\mathrm{d}}}, \ \max_{\theta \in \{\theta_1, \ldots, \theta_{\ell-1}\}} \frac{u_\theta^{\mathrm{a}}(\pi(\theta)) - r_i^\theta}{p_i^\theta - r_i^\theta} \right\}$.

4. If $\mathrm{EoP}(\pi) \geq \xi$, return $\pi$ as a satisfying policy; Otherwise, claim that no such policy exists.

---

**Theorem 10.** *In time polynomial in $m$, $n$, and $|\Theta|$, Algorithm 1 either outputs a policy $\pi$ with $\mathrm{EoP}(\pi) \geq \xi$, or decides correctly that no such policy exists. The policy generated is IC.*

*Proof.* The polynomial runtime is readily seen. By Lemma 9, $\pi(\theta_\ell)$ generated in Step 3 must be a feasible outcome, so $\pi$ is a feasible policy. When no feasible policy can achieve EoP $\xi$, we have $\mathrm{EoP}(\pi) < \xi$ and Algorithm 1 will decide correctly in Step 4 that no satisfying policy exists.

Suppose that there exists a policy $\pi^*$, $\mathrm{EoP}(\pi^*) \geq \xi$. Let $\tilde{\pi}^0 = \pi^*$, and for each $\ell = 1, \ldots, \lambda$, we iteratively construct a policy $\tilde{\pi}^\ell$ by replacing $\tilde{\pi}^{\ell-1}(\theta_\ell)$ in policy $\tilde{\pi}^{\ell-1}$ with $\pi(\theta_\ell)$ generated in Step 3; thus, $\tilde{\pi}^\lambda = \pi$. Trivially, $\tilde{\pi}^0$ is 0-compatible as is any feasible policy. Applying Lemma 9 iteratively, we can also conclude that $\tilde{\pi}^\ell$ is $\ell$-compatible and $\mathrm{EoP}(\tilde{\pi}^\ell) \geq \xi$ for every $\ell$; in particular, $\tilde{\pi}^\lambda$ is $\lambda$-compatible and $\mathrm{EoP}(\pi^\lambda) \geq \xi$. Algorithm 1 outputs $\pi = \tilde{\pi}^\lambda$ as a satisfying policy. $\qquad\square$

## 5.2 Beyond Finite Attacker Types

The above approach only applies to a finite type set, we present a heuristic approach to deal with a continuous or even unknown type set. The approach is inspired by the *quantal response (QR)* model that is developed to study bounded rationality of human players [13]. In a QR equilibrium, players are assumed to play not only their optimal pure strategy but also every other strategy with a probability positively related to the utility the player gets from playing that strategy.

The QR policy imitates the irrational behavior in a QR equilibrium. Recall that in an SSE, a rational defender commits to an optimal strategy $\mathbf{c}$ and induces a type-$\theta$ attacker to choose a response $i^* \in \mathrm{BR}_\theta(\mathbf{c})$ that maximizes the defender's utility. The QR policy, however, induces the attacker's tie-breaking choice in an "irrational" way. It induces the attacker to choose not only $i^*$, but also *every* target in $\mathrm{BR}_\theta(\mathbf{c})$ with some probability; the probability a target being chosen is positively related to the defender's utility when this target is attacked. The idea is to add some uncertainty in the outcome, so that the attacker cannot benefit from being induced to choose a particular response with certainty, which is crucial for the success of his manipulation. This also encourages truthful report to some extent: a truthful attacker, who reports his true type $\theta$, is indifferent of which response he is induced to choose in $\mathrm{BR}_\theta(\mathbf{c})$ and hence immune to such uncertainty. The QR policy is as follows.

**Definition 11 (QR policy).** For each type $\theta$, let $\hat{\mathbf{c}}^\theta$ be the defender strategy in an SSE on attacker type $\theta$. A QR policy $\pi^{\mathrm{QR}}$ maps a report $\theta$ to a distribution $\sigma$ over outcomes in $\{(\hat{\mathbf{c}}^\theta, i) : i \in \mathrm{BR}_\theta(\hat{\mathbf{c}}^\theta)\}$; the probability $\sigma(i)$ of each outcome $(\hat{\mathbf{c}}^\theta, i)$ is $\sigma(i) = \frac{f_i}{\sum_{j \in \mathrm{BR}_\theta(\hat{\mathbf{c}}^\theta)} f_j}$, where $f_j = e^{\varphi \cdot u^{\mathrm{d}}(\hat{\mathbf{c}}^\theta, j)}$ for each $j \in T$, with $\varphi > 0$ being a parameter that represents a player's rationality level in the QR model.[3]

A defender who uses $\pi^{\mathrm{QR}}$ then samples an outcome from $\pi^{\mathrm{QR}}(\theta)$ to implement when $\theta$ is reported. The players are now concerned with their expected utility over the outcome distribution, e.g., for the defender: $u^{\mathrm{d}}(\pi^{\mathrm{QR}}(\theta)) = \sum_{i \in \mathrm{BR}_\theta(\hat{\mathbf{c}}^\theta)} \sigma(i) \cdot u^{\mathrm{d}}(\hat{\mathbf{c}}^\theta, i)$. The EoP can also be redefined accordingly. Since $\hat{\mathbf{c}}^\theta$ and $\sigma$ are independent of other types in $\Theta$, the QR policy can be implemented on-the-fly for the type reported, and is thus able to handle infinite or unknown type sets.

Intuitively, the QR policy strikes a balance between two unaligned aspects of playing against attacker manipulation: 1) it adds some uncertainty to discourage attacker manipulation; meanwhile, 2) the softmax function that defines $\sigma$ loosely strings the induced attacker response to the optimal one for

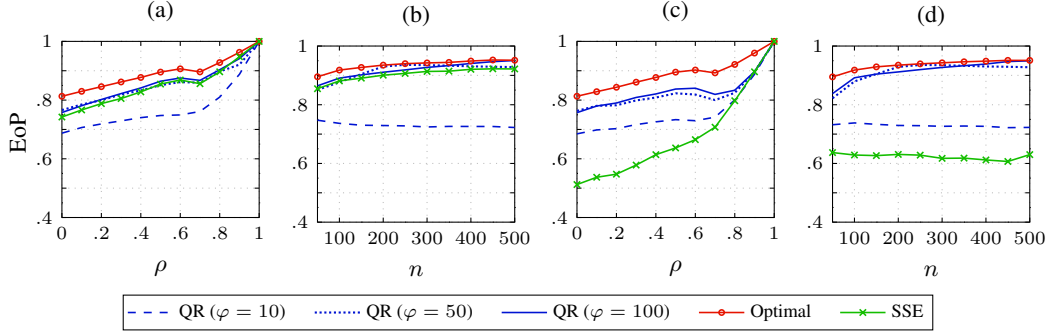

Figure 1: Comparison of the EoP. In (a), results are obtained with other parameters set to $\lambda = 100$, $m = 10$, and $n = 50$; and in (b) with $m = n/5$, $\rho = 0.5$, and $\lambda = 100$. Figures (c) and (d) repeat (a) and (b), respectively, with the difference that the zero-sum attacker type is always included in $\Theta$ in the experiments.

the defender, so that the cost of achieving 1) is kept away from being too high. We empirically evaluate the performance of the QR policy in randomly generated games in the next section.

## 6   Empirical Evaluation

In our evaluations, attacker types are randomly generated using the covariance model [15], with a parameter $\rho \in [0, 1]$ to control the closeness of the generated game to a zero-sum game. That is, we shift each payoff parameter $x$ towards the corresponding one $y$ of a zero-sum attacker type, letting $x \leftarrow (1 - \rho) \cdot x + \rho \cdot y$. Thus, when $\rho = 1$ the game generated is exactly zero-sum, and when $\rho = 0$ all payoffs are generated uniformly at random. All evaluations are conducted on finite type sets. This also simulates situations with an unknown (but finite) type set, though situations with infinite type sets requires more advanced approaches (in this case, it is unclear to us how to compute the optimal attacker report in response to the QR policy). All results shown are the average of at least 50 runs.

We compare the EoP achieved by our optimal and heuristic policies, using the SSE policy as a benchmark (i.e., the situation when attacker manipulation is ignored). The first set of results, Figure 1 (a) and (b), shows the variance of the EoP with respect to $\rho$ and the size of the game. Except for the QR policy with $\varphi = 10$, performance of all other policies is very close to each other, though there is a discernable gap between the optimal policy and the SSE policy. In general, in these results, the loss due to ignoring attacker manipulation appears to be very marginal.

A more interesting set of results is shown (c) and (d), in which we slightly tweak the randomly generated type set, by always adding a zero-sum attacker type in it. This small change leads to a very different pattern in the results. There is a wide gap between the optimal and the SSE policies, and the QR policies normally rest in between them, exhibiting good performance as well. The results corroborate our theoretical analysis, that all attacker types will be incentivized to report the zero-sum type when they are allowed to, which undermines the performance of the SSE policy significantly. The optimal policy, however, is able to achieve very high EoP, sometimes close to recovering the defender's utility in the truthful situation (EoP $= 1$).

## 7   Conclusion

In this paper, we investigate manipulation of algorithms that are designed to learn the optimal strategy to commit to in Stackelberg security games, and aim at remedying the overoptimistic assumption of a truthful attacker adopted by these algorithms. We propose exact and heuristic approaches to reduce the loss due to manipulation. The effectiveness of our approaches are evaluated both theoretically and empirically. One promising direction for future work is to look at similar problems in other variants of Stackelberg games, where our framework and approaches may apply.

**Acknowledgments**

Jiarui Gan was supported by the EPSRC International Doctoral Scholars Grant EP/N509711/1.

## Footnotes

[1] See [20] and [21] (Chapter 8) for more discussion about the SSG and the solution concepts.

[2]We also specify an attacker response in the output of a policy in order to deal with the tie-breaking issue explicitly. Same as in SSEs, the defender can induce specific attacker responses through infinitesimal deviations.

[3]When $\varphi \to 0$, a player behaves completely irrationally, playing each strategy uniformly at random; when $\varphi \to +\infty$, a player becomes perfectly rational, choosing the optimal strategy with certainty.

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
