[Supplementary Material]

# Appendix

## A    Optimal Attacker Report

**Lemma 12.** *Suppose $(\hat{\mathbf{c}}, \hat{i})$ is an SSE on attacker type $\theta$. The following holds:*

    *(i) If $\hat{c}_i < 1$ for all $i \in T$, then $\{i \in T : \hat{c}_i > 0\} \subseteq \mathrm{BR}_\theta(\hat{\mathbf{c}})$ and $\sum_{i \in T} \hat{c}_i = m$.*

    *(ii) If $\hat{c}_i = 1$ for some $i \in T$, then there exists $j \in \mathrm{BR}_\theta(\hat{\mathbf{c}})$ such that $\hat{c}_j = 1$.*

*Proof.* Since $(\hat{\mathbf{c}}, \hat{i})$ is an SSE, by definition, we have $\hat{i} \in \mathrm{BR}_\theta(\hat{\mathbf{c}})$, and $u^{\mathrm{d}}(\hat{\mathbf{c}}, \hat{i}) \geq u^{\mathrm{d}}(\mathbf{c}, i)$ for all $\mathbf{c} \in \mathcal{C}, i \in \mathrm{BR}_\theta(\mathbf{c})$. We claim that $\hat{\mathbf{c}}$ is the optimal solution to the following linear program:

$$\max_{\mathbf{c}} \quad u^{\mathrm{d}}(\mathbf{c}, \hat{i}) \tag{4}$$
$$\text{s.t.} \quad u^{\mathrm{a}}_\theta(\mathbf{c}, i) \leq u^{\mathrm{a}}_\theta(\mathbf{c}, \hat{i}) \qquad \text{for all } i \in T \setminus \{\hat{i}\} \tag{4a}$$
$$\sum_{i \in T} c_i \leq m \tag{4b}$$
$$0 \leq c_i \leq 1 \qquad \text{for all } i \in T \tag{4c}$$

To see this, note that, first $\hat{\mathbf{c}}$ is a feasible solution because it satisfies all the constraints above: (4a) is equivalent to $\hat{i} \in \mathrm{BR}_\theta(\mathbf{c})$, and (4b) and (4c) combined are equivalent to $\mathbf{c} \in \mathcal{C}$. Second, $\hat{\mathbf{c}}$ is optimal because if it is not, there would exist another feasible solution $\mathbf{z} \neq \hat{\mathbf{c}}$, such that $u^{\mathrm{d}}(\mathbf{z}, \hat{i}) > u^{\mathrm{d}}(\hat{\mathbf{c}}, \hat{i})$; this contradicts the assumption that $(\hat{\mathbf{c}}, \hat{i})$ forms an SSE.

By the Karush-Kuhn-Tucker (KKT) conditions, $\hat{\mathbf{c}}$ is an optimal solution only if there exists constants (i.e., KKT multipliers) $\alpha_i, \beta, \gamma_i$ and $\delta_i$, each corresponding to an inequality constraint in (4a)–(4c), such that for all $i \in T$ (let $w^{\mathrm{a}}_i = p^\theta_i - r^\theta_i$ and $w^{\mathrm{d}}_i = r^{\mathrm{d}}_i - p^{\mathrm{d}}_i$ for each $i$ below):

$$\begin{cases} -w^{\mathrm{a}}_i \cdot \alpha_i \quad -\beta + \gamma_i - \delta_i = 0 \quad \text{for all } i \neq \hat{i} \\ w^{\mathrm{d}}_{\hat{i}} + w^{\mathrm{a}}_{\hat{i}} \cdot \sum_{i \neq \hat{i}} \alpha_i - \beta + \gamma_{\hat{i}} - \delta_{\hat{i}} = 0 \end{cases} \qquad \text{(by stationarity}^4\text{)} \tag{5}$$

$$\alpha_i, \beta, \gamma_i, \delta_i \geq 0 \qquad \text{(by dual feasibility)} \tag{6}$$

$$\begin{cases} \alpha_i \cdot \left( u^{\mathrm{a}}_\theta(\hat{\mathbf{c}}, i) - u^{\mathrm{a}}_\theta(\hat{\mathbf{c}}, \hat{i}) \right) = 0 \\ \beta \cdot \left( \sum_{j \in T} \hat{c}_j - m \right) = 0 \\ \gamma_i \cdot \hat{c}_i = 0 \\ \delta_i \cdot (\hat{c}_i - 1) = 0 \end{cases} \qquad \text{(by complementary slackness}^5\text{)} \tag{7}$$

Now we show (i) and (ii) in the statement of the lemma separately.

**Part (i).** Since $\hat{c}_i < 1$ for all $i \in T$, we have $\delta_i = 0$ for all $i \in T$ by the last equation in (7). Suppose towards a contradiction that $\sum_{i \in T} \hat{c}_i < m$. We would have $\beta = 0$ by the second equation in (7); and further, by the second equation in (5), $w^{\mathrm{d}}_{\hat{i}} + w^{\mathrm{a}}_{\hat{i}} \cdot \sum_{i \neq \hat{i}} \alpha_i + \gamma_{\hat{i}} = 0$, which is a contradiction as $w^{\mathrm{d}}_{\hat{i}} = r^{\mathrm{d}}_{\hat{i}} - p^{\mathrm{d}}_{\hat{i}} > 0$, $w^{\mathrm{a}}_{\hat{i}} = r^{\mathrm{a}}_{\hat{i}} - p^{\mathrm{a}}_{\hat{i}} > 0$, and $\alpha_i, \gamma_i \geq 0$ for all $i$ by (5). Thus, $\beta > 0$ and $\sum_{i \in T} \hat{c}_i = m$. Similarly, if we suppose $\hat{c}_t > 0$ for some $t \in T$, but $t \notin \mathrm{BR}_\theta(\hat{\mathbf{c}})$, we would have $u^{\mathrm{a}}_\theta(\hat{\mathbf{c}}, t) < \max_{i \in T} u^{\mathrm{a}}_\theta(\hat{\mathbf{c}}, i) = u^{\mathrm{a}}_\theta(\hat{\mathbf{c}}, \hat{i})$. Thus, $\gamma_t = 0$ and $\alpha_t = 0$ by the third and the first equations in (7), and then $\beta + \delta_t = 0$ by the first equation in (5) (note that $t \neq \hat{i}$ since $\hat{i} \in \mathrm{BR}_\theta(\hat{\mathbf{c}})$), which contradicts $\beta > 0$ and $\delta_i = 0$ for all $i \in T$ as we show above.

**Part (ii).** Suppose that $\hat{c}_i = 1$ for some $i \in T$, but $\hat{c}_j < 1$ for all $j \in \mathrm{BR}_\theta(\mathbf{c})$ (in particular, $\hat{c}_{\hat{i}} < 1$). Thus, $i \notin \mathrm{BR}_\theta(\hat{\mathbf{c}})$, so $u_\theta^{\mathrm{a}}(\hat{\mathbf{c}}, i) < \max_{t \in T} u_\theta^{\mathrm{a}}(\hat{\mathbf{c}}, t) = u_\theta^{\mathrm{a}}(\hat{\mathbf{c}}, \hat{i})$. We have $\gamma_i = 0$ and $\alpha_i = 0$ by (7), which then implies $\beta + \delta_i = 0$ by (5); thus, $\beta = 0$. In addition, $\hat{c}_{\hat{i}} < 1$ implies $\delta_{\hat{i}} = 0$ by (7). Again, by (5), we end up with the contradiction that $w_{\hat{i}}^{\mathrm{d}} + w_{\hat{i}}^{\mathrm{a}} \cdot \sum_{i \neq \hat{i}} \alpha_i + \gamma_{\hat{i}} = 0$. $\qquad\square$

**Lemma 13.** *Suppose $\bar{\mathbf{c}}$ is a maximin strategy of the defender, i.e., $\bar{\mathbf{c}} \in \arg\max_{\mathbf{c} \in \mathcal{C}} \min_{i \in T} u^{\mathrm{d}}(\mathbf{c}, i)$. Then $(\bar{\mathbf{c}}, i)$ forms an SSE in a zero-sum game for any $i \in \mathrm{BR}_\beta(\bar{\mathbf{c}})$, where $\beta = (-\mathbf{p}^{\mathrm{d}}, -\mathbf{r}^{\mathrm{d}})$.*

*Proof.* Suppose $(\bar{\mathbf{c}}, i)$, $i \in \mathrm{BR}_\beta(\bar{\mathbf{c}})$, is not an SSE. Thus, there exists $\mathbf{z} \in \mathcal{C}$ and $t \in \mathrm{BR}_\beta(\mathbf{z})$, such that $u^{\mathrm{d}}(\mathbf{z}, t) > u^{\mathrm{d}}(\bar{\mathbf{c}}, i)$; equivalently, $u_\beta^{\mathrm{a}}(\mathbf{z}, t) < u_\beta^{\mathrm{a}}(\bar{\mathbf{c}}, i)$ as $\beta$ makes the game zero-sum. Since $i \in \mathrm{BR}_\beta(\bar{\mathbf{c}})$ and $t \in \mathrm{BR}_\beta(\mathbf{z})$ are the attacker's best responses, we have $u_\beta^{\mathrm{a}}(\bar{\mathbf{c}}, i) = \max_{j \in T} u_\beta^{\mathrm{a}}(\bar{\mathbf{c}}, j)$ and $u_\beta^{\mathrm{a}}(\mathbf{z}, t) = \max_{j \in T} u_\beta^{\mathrm{a}}(\mathbf{z}, j)$; thus, $\max_{j \in T} u_\beta^{\mathrm{a}}(\mathbf{z}, j) < \max_{j \in T} u_\beta^{\mathrm{a}}(\bar{\mathbf{c}}, j)$. This leads to the following contradiction:

$$\max_{\mathbf{c} \in \mathcal{C}} \min_{j \in T} u^{\mathrm{d}}(\mathbf{c}, j) = \min_{j \in T} u^{\mathrm{d}}(\bar{\mathbf{c}}, j) = -\max_{j \in T} u_\beta^{\mathrm{a}}(\bar{\mathbf{c}}, j) < -\max_{j \in T} u_\beta^{\mathrm{a}}(\mathbf{z}, j) = \min_{j \in T} u^{\mathrm{d}}(\mathbf{z}, j). \quad\square$$

**Corollary 14.** *Suppose $\bar{\mathbf{c}}$ is a maximin strategy of the defender and $\bar{\mathbf{c}}$ is fully mixed, i.e., $0 < \bar{c}_i < 1$ for all $i \in T$. Then $(\bar{\mathbf{c}}, i)$ forms an SSE in a zero-sum game for all $i \in T$, and $u^{\mathrm{d}}(\bar{\mathbf{c}}, i) = \min_{j \in T} u^{\mathrm{d}}(\bar{\mathbf{c}}, j)$ for all $i \in T$.*

*Proof.* By Lemma 13, $(\bar{\mathbf{c}}, i)$ forms an SSE on attacker type $\beta = (-\mathbf{p}^{\mathrm{d}}, -\mathbf{r}^{\mathrm{d}})$ with any $i \in \mathrm{BR}_\beta(\bar{\mathbf{c}})$. Since $\bar{\mathbf{c}}$ is fully mixed, $T = \{i \in T : \bar{c}_i > 0\}$, and by Lemma 12 (i), $T \subseteq \mathrm{BR}_\beta(\bar{\mathbf{c}}) \subseteq T$. Thus, $\mathrm{BR}_\beta(\bar{\mathbf{c}}) = T$, so $(\bar{\mathbf{c}}, i)$ forms an SSE on attacker type $\beta$ (which makes the game zero-sum) with any $i \in T$; we have $u_\beta^{\mathrm{a}}(\bar{\mathbf{c}}, i) = \max_{j \in T} u_\beta^{\mathrm{a}}(\bar{\mathbf{c}}, j)$. It follows that $u^{\mathrm{d}}(\bar{\mathbf{c}}, i) = -u_\beta^{\mathrm{a}}(\bar{\mathbf{c}}, i) = -\max_{j \in T} u_\beta^{\mathrm{a}}(\bar{\mathbf{c}}, j) = \min_{j \in T} u^{\mathrm{d}}(\bar{\mathbf{c}}, j)$. $\qquad\square$

**Lemma 15.** *Suppose $\bar{\mathbf{c}}$ is a maximin strategy of the defender and $\bar{\mathbf{c}}$ is fully mixed, i.e., $0 < \bar{c}_i < 1$ for all $i \in T$. Then $\bar{\mathbf{c}}$ is the only maximin strategy of the defender.*

*Proof.* Suppose $\mathbf{z} = \arg\max_{\mathbf{c} \in \mathcal{C}} \min_{i \in T} u^{\mathrm{d}}(\mathbf{c}, i)$ is a maximin strategy and $\mathbf{z} \neq \bar{\mathbf{c}}$. Thus, either: (i) $z_i \geq \bar{c}_i$ for all $i \in T$, and this is strictly satisfied by some $i$; or (ii) $z_i < \bar{c}_i$ for some $i \in T$. We show either of them leads to a contradiction.

Since $\bar{\mathbf{c}}$ is a maximin strategy, it is also an SSE defender strategy in a zero-sum game by Lemma 13; and by Lemma 12 (i), $\sum_{i \in T} \bar{c}_i = m$. Thus, in Case (i), it follows immediately that $\sum_{i \in T} z_i > \sum_{i \in T} \bar{c}_i = m$, which contradicts $\mathbf{z} \in \mathcal{C}$. In Case (ii), we have $u^{\mathrm{d}}(\mathbf{z}, i) < u^{\mathrm{d}}(\bar{\mathbf{c}}, i)$ by monotonicity of $u^{\mathrm{d}}(\cdot, i)$, which implies $\min_{j \in T} u^{\mathrm{d}}(\mathbf{z}, j) \leq u^{\mathrm{d}}(\mathbf{z}, i) < u^{\mathrm{d}}(\bar{\mathbf{c}}, i) = \min_{j \in T} u^{\mathrm{d}}(\bar{\mathbf{c}}, j)$, where the last equality follows by Corollary 14. This contradicts the assumption that $\mathbf{z}$ is a maximin strategy. $\qquad\square$

## A.1 Proof of Theorem 3

*Proof.* Let $\bar{\mathbf{c}}$ be a maximin strategy of the defender and $\bar{u}$ be the corresponding maximin utility, i.e., $\bar{\mathbf{c}} \in \arg\max_{\mathbf{c} \in \mathcal{C}} \min_{i \in T} u^{\mathrm{d}}(\mathbf{c}, i)$ and $\bar{u} = \min_{i \in T} u^{\mathrm{d}}(\bar{\mathbf{c}}, i)$. Consider the following solution $(\beta, \mathbf{z}, t)$:

- $z_i = \max\left\{0, \frac{\bar{u} - p_i^{\mathrm{d}}}{r_i^{\mathrm{d}} - p_i^{\mathrm{d}}}\right\}$ for all $i \in T$; $\qquad\qquad\qquad\qquad\qquad\qquad\qquad$ (8)

- $t \in \mathrm{BR}_\theta(\mathbf{z})$ is an arbitrary best response of a type-$\theta$ attacker; $\qquad\qquad\qquad$ (9)

- $\beta = (\mathbf{r}, \mathbf{p})$, where $r_i = \begin{cases} -p_i^{\mathrm{d}}, & \text{if } i \neq t \\ -\min\{p_t^{\mathrm{d}}, \bar{u}\}, & \text{if } i = t \end{cases}$, and $p_i = -r_i^{\mathrm{d}}$ for all $i \in T$. $\qquad$ (10)

We show that (i) $\mathbf{z}$ is a maximin defender strategy, (ii) $(\beta, \mathbf{z}, t)$ is a feasible solution of Program (3) and (iii) it is optimal.

We first focus on the case when $p_t^{\mathrm{d}} \leq \bar{u}$, and will show how the proof can be modified to show the same results when $p_t^{\mathrm{d}} > \bar{u}$. When $p_t^{\mathrm{d}} \leq \bar{u}$ we have $(r_i, p_i) = (-p_i^{\mathrm{d}}, -r_i^{\mathrm{d}})$ for all $i \in T$ by the specification in (10), so for any $\mathbf{c} \in \mathcal{C}$,

$$u_\beta^{\mathrm{a}}(\mathbf{c}, i) = -u^{\mathrm{d}}(\mathbf{c}, i). \qquad\qquad\qquad\qquad\qquad\qquad\qquad (11)$$

Now we show (i)–(iii).

**(i) Maximin.** For all $i \in T$, since $z_i = \max\left\{0, \frac{\overline{u}-p_i^{\mathrm{d}}}{r_i^{\mathrm{d}}-p_i^{\mathrm{d}}}\right\} \geq \frac{\overline{u}-p_i^{\mathrm{d}}}{r_i^{\mathrm{d}}-p_i^{\mathrm{d}}}$, we have

$$u^{\mathrm{d}}(\mathbf{z}, i) \geq u^{\mathrm{d}}\left(\frac{\overline{u}-p_i^{\mathrm{d}}}{r_i^{\mathrm{d}}-p_i^{\mathrm{d}}}, i\right) = \frac{\overline{u}-p_i^{\mathrm{d}}}{r_i^{\mathrm{d}}-p_i^{\mathrm{d}}} \cdot r_i^{\mathrm{d}} + \left(1 - \frac{\overline{u}-p_i^{\mathrm{d}}}{r_i^{\mathrm{d}}-p_i^{\mathrm{d}}}\right) \cdot p_i^{\mathrm{d}} = \overline{u}. \tag{12}$$

It follows that

$$\min_{i \in T} u^{\mathrm{d}}(\mathbf{z}, i) \geq \overline{u} \geq \min_{i \in T} u^{\mathrm{d}}(\mathbf{c}, i)$$

for all $\mathbf{c} \in \mathcal{C}$, so $\mathbf{z}$ is indeed a maximin strategy. We still need to make sure that $\mathbf{z}$ is feasible, i.e., $\mathbf{z} \in \mathcal{C}$. Observe that when $z_i > 0$, (12) becomes an equality, so we have $u^{\mathrm{d}}(\mathbf{z}, i) = \overline{u} \leq u^{\mathrm{d}}(\overline{\mathbf{c}}, i)$ which implies $z_i \leq \overline{c}_i$ by monotonicity. Thus, $\sum_{i \in T} z_i \leq \sum_{i \in T} \overline{c}_i \leq m$. It remains to show that $0 \leq z_i \leq 1$ for all $i$. Trivially, by (8), $z_i \geq 0$ for all $i$. To see that $z_i \leq 1$, it suffices to show that $\frac{\overline{u}-p_i^{\mathrm{d}}}{r_i^{\mathrm{d}}-p_i^{\mathrm{d}}} \leq 1$. Indeed, this holds as $\overline{u} = \min_{j \in T} u^{\mathrm{d}}(\overline{\mathbf{c}}, j) \leq u^{\mathrm{d}}(\overline{\mathbf{c}}, i) \leq r_i^{\mathrm{d}}$.

**(ii) Feasibility** We show that $(\beta, \mathbf{z}, t)$ satisfies all the constraints of Program (3). Clearly, (3b) is satisfied because $r_i \geq -p_i^{\mathrm{d}} > -r_i^{\mathrm{d}} = p_i$ for all $i \in T$ by (10). To see that it also satisfies (3a), first observe that when $p_t^{\mathrm{d}} \leq \overline{u}$ we have

$$u_\beta^{\mathrm{a}}(\mathbf{z}, t) = u_\beta^{\mathrm{a}}\left(\frac{\overline{u}-p_t^{\mathrm{d}}}{r_t^{\mathrm{d}}-p_t^{\mathrm{d}}}, t\right) = \left(1 - \frac{\overline{u}-p_t^{\mathrm{d}}}{r_t^{\mathrm{d}}-p_t^{\mathrm{d}}}\right) \cdot (-p_t^{\mathrm{d}}) + \frac{\overline{u}-p_t^{\mathrm{d}}}{r_t^{\mathrm{d}}-p_t^{\mathrm{d}}} \cdot (-r_t^{\mathrm{d}}) = -\overline{u}. \tag{13}$$

Combining this with (11) and (12) gives, for all $i$,

$$u_\beta^{\mathrm{a}}(\mathbf{z}, t) = -\overline{u} \geq -u^{\mathrm{d}}(\mathbf{z}, i) = u_\beta^{\mathrm{a}}(\mathbf{z}, i). \tag{14}$$

Thus, $t \in \mathrm{BR}_\beta(\mathbf{z})$.

Now that $t \in \mathrm{BR}_\beta(\mathbf{z})$ and in (i) we have shown that $\mathbf{z} \in \mathcal{C}$, if we suppose (3a) is not satisfied, we would have $u^{\mathrm{d}}(\mathbf{z}', t') > u^{\mathrm{d}}(\mathbf{z}, t)$ for some $\mathbf{z}' \in \mathcal{C}$ and $t' \in \mathrm{BR}_\beta(\mathbf{z}')$. Applying (11), we find the following for all $i$:

$$u^{\mathrm{d}}(\mathbf{z}', i) = -u_\beta^{\mathrm{a}}(\mathbf{z}', i) \geq -u_\beta^{\mathrm{a}}(\mathbf{z}', t') = u^{\mathrm{d}}(\mathbf{z}', t'),$$

where the inequality is due to the fact that $t' \in \mathrm{BR}_\beta(\mathbf{z}')$. Thus,

$$u^{\mathrm{d}}(\mathbf{z}', i) \geq u^{\mathrm{d}}(\mathbf{z}', t') > u^{\mathrm{d}}(\mathbf{z}, t) \geq \min_{i \in T} u^{\mathrm{d}}(\mathbf{z}, i) = \overline{u}. \tag{15}$$

It follows that

$$\min_{i \in T} u^{\mathrm{d}}(\mathbf{z}', i) > \overline{u} = \max_{\mathbf{c} \in \mathcal{C}} \min_{i \in T} u^{\mathrm{d}}(\mathbf{c}, i),$$

which is a contradiction given that $\mathbf{z}' \in \mathcal{C}$.

**(iii) Optimality.** Suppose that $(\beta, \mathbf{z}, t)$ is not optimal. Thus, there exists a feasible solution $(\beta', \mathbf{z}', t')$ such that $u_\theta^{\mathrm{a}}(\mathbf{z}', t') > u_\theta^{\mathrm{a}}(\mathbf{z}, t)$. By 9, $t \in \mathrm{BR}_\theta(\mathbf{z})$, so we have $u_\theta^{\mathrm{a}}(\mathbf{z}, t') \leq u_\theta^{\mathrm{a}}(\mathbf{z}, t) < u_\theta^{\mathrm{a}}(\mathbf{z}', t')$, which implies $z_{t'}' < z_{t'}$ by monotonicity. Since it is defined $z_{t'} = \max\left\{0, \frac{\overline{u}-p_{t'}^{\mathrm{d}}}{r_{t'}^{\mathrm{d}}-p_{t'}^{\mathrm{d}}}\right\}$, now that $z_{t'} > z_{t'}' \geq 0$, it must be that $z_{t'}' < z_{t'} = \frac{\overline{u}-p_{t'}^{\mathrm{d}}}{r_{t'}^{\mathrm{d}}-p_{t'}^{\mathrm{d}}}$. Substituting this into the defender's utility function gives

$$u^{\mathrm{d}}(\mathbf{z}', t') < u^{\mathrm{d}}\left(\frac{\overline{u}-p_{t'}^{\mathrm{d}}}{r_{t'}^{\mathrm{d}}-p_{t'}^{\mathrm{d}}}, t'\right) = \frac{\overline{u}-p_{t'}^{\mathrm{d}}}{r_{t'}^{\mathrm{d}}-p_{t'}^{\mathrm{d}}} \cdot r_i^{\mathrm{d}} + \left(1 - \frac{\overline{u}-p_{t'}^{\mathrm{d}}}{r_{t'}^{\mathrm{d}}-p_{t'}^{\mathrm{d}}}\right) \cdot p_i^{\mathrm{d}}$$

$$= \overline{u} = \max_{\mathbf{c} \in \mathcal{C}} \min_{i \in T} u^{\mathrm{d}}(\mathbf{c}, i) \leq \max_{\mathbf{c} \in \mathcal{C}, i \in \mathrm{BR}_\beta(\mathbf{c})} u^{\mathrm{d}}(\mathbf{c}, i).$$

Thus, $(\beta', \mathbf{z}', t')$ violates (3a), contradicting the assumption that $(\beta', \mathbf{z}', t')$ is a feasible solution.

It remains to deal with the case when $p_t^{\mathrm{d}} > \overline{u}$. The only difference in this case is that now $r_t = -\overline{u} > -p_t^{\mathrm{d}}$ by (10), so (11) only holds for $i \neq t$. In our proof above, the arguments that rely on the assumption that $p_t^{\mathrm{d}} \leq \overline{u}$ and (11) are the equations in (13), (14), and (15), where the latter two now only hold for all $i \neq t$. However, observe the following:

- When $p_t^{\mathrm{d}} > \overline{u}$, we have $\frac{\overline{u} - p_t^{\mathrm{d}}}{r_t^{\mathrm{d}} - p_t^{\mathrm{d}}} < 0$ and thus $z_t = 0$ by (8). It follows that $u_\beta^{\mathrm{a}}(\mathbf{z}, t) = u_\beta^{\mathrm{a}}(0, t) = r_t = -\overline{u}$, so (13) holds as well.

- (14) holds trivially for $i = t$.

- When $p_t^{\mathrm{d}} > \overline{u}$, we have $u^{\mathrm{d}}(\mathbf{z}', t) \geq p_t^{\mathrm{d}} > \overline{u}$, thus establishing (15) for $i = t$ as well.

Therefore, (i)–(iii) hold when $p_t^{\mathrm{d}} > \overline{u}$ and the proof is completed. $\qquad\square$

## A.2 Proof of Theorem 4

*Proof.* In Theorem 3, we have shown the existence of an optimal solution containing a maximin strategy of the defender. By Lemma 15, $\overline{\mathbf{c}}$ is the only maximin strategy if it is fully mixed. Thus, there exits an optimal solution that contains $\overline{\mathbf{c}}$. We fix $\overline{\mathbf{c}}$ in Program (3), and show that in the resultant program, an optimal solution $(\beta', t')$ is such that $\beta' = (-\mathbf{p}^{\mathrm{d}}, -\mathbf{r}^{\mathrm{d}})$ and $t' \in \arg\max_{i \in T} u_\theta^{\mathrm{a}}(\overline{\mathbf{c}}, i)$. In fact, now that $\overline{\mathbf{c}}$ is fixed, the optimality of $(\beta', t')$ follows directly from the specification $t' \in \arg\max_{i \in T} u_\theta^{\mathrm{a}}(\overline{\mathbf{c}}, i)$. It remains to show that $(\beta', t')$ is feasible. Since the game with attacker type $\beta'$ is a zero-sum game, by Lemma 13, $(\overline{\mathbf{c}}, t')$ forms an SSE on type $\beta'$. Thus, Constraint (3a) is satisfied, and $(\beta', t')$ is feasible. Therefore, $(\beta', \overline{\mathbf{c}}, t')$ is an optimal solution to Program (3).

Now we show the first part of the theorem, i.e., the uniqueness of the induced defender strategy. Suppose for a contradiction that there exists an SSE $(\hat{\mathbf{c}}, \hat{i})$ on attacker type $\beta$, such that $\hat{\mathbf{c}} \neq \overline{\mathbf{c}}$. Consider the two possibilities under this condition.

**Case (i).** $\hat{c}_i \leq \overline{c}_i$ for all $i \in T$, and this is strictly satisfied for some $i$. It follows that $\sum_{i \in T} \hat{c}_i < \sum_{i \in T} \overline{c}_i \leq m$, which contradicts Lemma 12 (i).

**Case (ii).** $\hat{c}_j > \overline{c}_j$ for some $j \in T$. If it is also the case that $\hat{c}_i < 1$ for all $i \in T$, by Lemma 12 (i), we have $j \in \mathrm{BR}_\beta(\hat{\mathbf{c}})$; If otherwise $\hat{c}_i = 1$ for some $i \in T$, by Lemma 12 (ii), there exists $j' \in \mathrm{BR}_\beta(\hat{\mathbf{c}})$ such that $\hat{c}_{j'} = 1 > \overline{c}_{j'}$. In both cases, we find some $j \in \mathrm{BR}_\beta(\hat{\mathbf{c}})$ such that $\hat{c}_j > \overline{c}_j$; hence, $u^{\mathrm{d}}(\hat{\mathbf{c}}, j) > u^{\mathrm{d}}(\overline{\mathbf{c}}, j)$ by monotonicity of $u^{\mathrm{d}}(\cdot, j)$. We have

$$u^{\mathrm{d}}(\hat{\mathbf{c}}, \hat{i}) = \max_{i \in \mathrm{BR}_\beta(\hat{\mathbf{c}})} u^{\mathrm{d}}(\hat{\mathbf{c}}, i) \geq u^{\mathrm{d}}(\hat{\mathbf{c}}, j) > u^{\mathrm{d}}(\overline{\mathbf{c}}, j) = u^{\mathrm{d}}(\overline{\mathbf{c}}, \hat{i}),$$

where the last equality follows by Corollary 14. Thus, by the monotonicity, we have $\hat{c}_{\hat{i}} > \overline{c}_{\hat{i}}$ and, in turn, $u_\theta^{\mathrm{a}}(\hat{\mathbf{c}}, \hat{i}) < u_\theta^{\mathrm{a}}(\overline{\mathbf{c}}, \hat{i})$. This gives

$$u_\theta^{\mathrm{a}}(\hat{\mathbf{c}}, \hat{i}) < u_\theta^{\mathrm{a}}(\overline{\mathbf{c}}, \hat{i}) \leq \max_{i \in T} u_\theta^{\mathrm{a}}(\overline{\mathbf{c}}, i) = u_\theta^{\mathrm{a}}(\overline{\mathbf{c}}, t'),$$

so $(\beta', \overline{\mathbf{c}}, t')$ is a better solution than $(\beta, \mathbf{z}, t)$ and this contradicts the assumption in the statement of the theorem.

Both cases lead to contradictions. This completes the proof. $\qquad\square$

# B The EoP Measure

## B.1 Proof of Proposition 5

*Proof.* Let $(\mathbf{z}, j) = \pi(\beta)$, which is the outcome a type-$\beta$ attacker would get if he reports truthfully. By definition, $j \in \mathrm{BR}_\beta(\mathbf{z})$. By Lemma 13, $\overline{\mathbf{c}}$ as the defender's maximin strategy is exactly her SSE strategy in a zero-sum game and $(\overline{\mathbf{c}}, t)$ forms an SSE for any $t \in \mathrm{BR}_\beta(\overline{\mathbf{c}})$. Thus, $u^{\mathrm{d}}(\mathbf{z}, j) \leq u^{\mathrm{d}}(\overline{\mathbf{c}}, t)$ and $u_\beta^{\mathrm{a}}(\overline{\mathbf{c}}, t) = \max_{i \in T} u_\beta^{\mathrm{a}}(\overline{\mathbf{c}}, i)$. Since $\beta$ makes the game zero-sum, $u_\beta^{\mathrm{a}}(\mathbf{c}, i) = u^{\mathrm{d}}(\mathbf{c}, i)$ for any $\mathbf{c}$ and $i$. It follows that

$$u_\beta^{\mathrm{a}}(\mathbf{z}, j) = -u^{\mathrm{d}}(\mathbf{z}, j) \geq -u^{\mathrm{d}}(\overline{\mathbf{c}}, t) = u_\beta^{\mathrm{a}}(\overline{\mathbf{c}}, t) = \max_{i \in T} u_\beta^{\mathrm{a}}(\overline{\mathbf{c}}, i) = -\min_{i \in T} u^{\mathrm{d}}(\overline{\mathbf{c}}, i) = -\overline{u}.$$

Now suppose towards a contradiction that $u^{\mathrm{d}}(\pi(\gamma)) > \overline{u}$. Then we have

$$u_\beta^{\mathrm{a}}(\pi(\gamma)) = -u^{\mathrm{d}}(\pi(\gamma)) < -\overline{u} \leq u_\beta^{\mathrm{a}}(\mathbf{z}, j) = u_\beta^{\mathrm{a}}(\pi(\beta)),$$

so the attacker would be strictly better-off reporting $\beta$. This contradicts the assumption that $\gamma$ is an optimal reporting strategy of a type-$\beta$ attacker. $\qquad\square$

## B.2 Proof of Proposition 7

*Proof.* Clearly, $\text{EoP}(\pi) \geq 0$ as the payoffs are shifted to be non-negative. We show that $\text{EoP}(\pi) \leq 1$.

Let $\hat{u}(\theta) = \max_{\mathbf{c} \in \mathcal{C}, i \in \text{BR}_\theta(\mathbf{c})} u^{\text{d}}(\mathbf{c}, i)$ denote the defender's utility in an SSE on attacker type $\theta$, and let $\beta$ be an attacker type that provides the best defender utility in an SSE, i.e., $\beta \in \arg \max_{\theta \in \Theta} \hat{u}(\theta)$. Consider the best reporting strategy $\gamma$ of a type-$\beta$ attacker in response to $\pi$. For the outcome $(\mathbf{z}, t) = \pi(\gamma)$ to be feasible, we must have $t \in \text{BR}_\gamma(\mathbf{z})$; thus, $u^{\text{d}}(\pi(\gamma)) = u^{\text{d}}(\mathbf{z}, t) \leq \max_{\mathbf{c} \in \mathcal{C}, i \in \text{BR}_\gamma(\mathbf{c})} u^{\text{d}}(\mathbf{c}, i) = \hat{u}(\gamma) \leq \hat{u}(\beta)$. It follows that

$$\text{EoP}(\pi) = \min_{\theta \in \Theta} \text{EoP}_\theta(\pi) \leq \text{EoP}_\beta(\pi) = \frac{u^{\text{d}}(\pi(\gamma))}{\hat{u}(\beta)} \leq 1. \qquad \square$$

## C Correctness of Algorithm 1

**Lemma 16.** *Let $(\hat{\mathbf{c}}, \hat{i})$ be an arbitrary SSE on an attacker type $\theta \in \Theta$. For any policy $\pi$, truthful report guarantees a type-$\theta$ attacker his SSE utility, i.e., $u^{\text{a}}_\theta(\pi(\theta)) \geq u^{\text{a}}_\theta(\hat{\mathbf{c}}, \hat{i})$.*

*Proof.* Suppose towards a contradiction that $u^{\text{a}}_\theta(\pi(\theta)) < u^{\text{a}}_\theta(\hat{\mathbf{c}}, \hat{i})$. Let $\pi(\theta) = (\mathbf{z}, t)$. For $\pi$ to be feasible, $t$ must be a best response of a type-$\theta$ attacker to $\mathbf{z}$. Thus, $u^{\text{a}}_\theta(\mathbf{z}, t) \geq u^{\text{a}}_\theta(\mathbf{z}, i)$ for all $i \in T$; in particular, $u^{\text{a}}_\theta(\mathbf{z}, t) \geq u^{\text{a}}_\theta(\mathbf{z}, \hat{i})$. We have

$$u^{\text{a}}_\theta(\hat{\mathbf{c}}, \hat{i}) > u^{\text{a}}_\theta(\pi(\theta)) = u^{\text{a}}_\theta(\mathbf{z}, t) \geq u^{\text{a}}_\theta(\mathbf{z}, \hat{i}).$$

Since $u^{\text{a}}_\theta(\mathbf{c}, \hat{i})$ changes continuously with respect to $c_{\hat{i}}$, the above inequality implies the existence of a number $\phi \in (\hat{c}_{\hat{i}}, z_{\hat{i}}]$, such that $u^{\text{a}}_\theta(\phi, \hat{i}) = u^{\text{a}}_\theta(\mathbf{z}, t)$.

Consider a defender strategy $\mathbf{z}'$ with $z'_{\hat{i}} = \phi$ and $z'_i = z_i$ for all $i \in T \setminus \{\hat{i}\}$. We have $0 \leq z_i \leq 1$ and $\sum_{i \in T} z'_i \leq \sum_{i \in T} z_i \leq m$, so $\mathbf{z}' \in \mathcal{C}$. In addition, $u^{\text{a}}_\theta(\mathbf{z}, i) = u^{\text{a}}_\theta(\mathbf{z}', i)$ for all $i \in T \setminus \{\hat{i}\}$. Thus,

$$u^{\text{a}}_\theta(\mathbf{z}', \hat{i}) = u^{\text{a}}_\theta(\phi, \hat{i}) = u^{\text{a}}_\theta(\mathbf{z}, t) \geq u^{\text{a}}_\theta(\mathbf{z}, i) = u^{\text{a}}_\theta(\mathbf{z}', i),$$

which means $\hat{i}$ is a best response of a type-$\theta$ attacker to $\mathbf{z}'$, i.e, $\hat{i} \in \text{BR}_\theta(\mathbf{z}')$. This gives rise to the following contradiction:

$$\max_{\mathbf{c} \in \mathcal{C}, i \in \text{BR}_\theta(\mathbf{c})} u^{\text{d}}(\mathbf{c}, i) \geq u^{\text{d}}(\mathbf{z}', \hat{i}) = u^{\text{d}}(\phi, \hat{i}) > u^{\text{d}}(\hat{\mathbf{c}}, \hat{i}) = \max_{\mathbf{c} \in \mathcal{C}, i \in \text{BR}_\theta(\mathbf{c})} u^{\text{d}}(\mathbf{c}, i),$$

where $u^{\text{d}}(\phi, \hat{i}) > u^{\text{d}}(\hat{\mathbf{c}}, \hat{i})$ since $\phi \in (\hat{c}_{\hat{i}}, z_{\hat{i}}]$. $\qquad \square$

### C.1 Proof of Lemma 9

*Proof.* We will write $\pi = (\mathbf{c}^\theta, i^\theta)_{\theta \in \Theta}$. Let $\beta \in \Theta$ be an optimal report of a type-$\theta_\ell$ attacker in response to $\pi$, i.e., $u^{\text{a}}_{\theta_\ell}(\pi(\beta)) \geq u^{\text{a}}_{\theta_\ell}(\pi(\beta'))$ for all $\beta' \in \Theta$. We first show a couple of useful observations.

***Claim 1.*** $u^{\text{a}}_{\theta_\ell}(\mathbf{h}, i^\beta) \geq u^{\text{a}}_{\theta_\ell}(\pi(\beta))$, i.e., a type-$\theta_\ell$ attacker (weakly) prefers outcome $(\mathbf{h}, i^\beta)$ to $\pi(\beta)$.

*Proof of Claim 1.* suppose towards a contradiction that $u^{\text{a}}_{\theta_\ell}(\mathbf{h}, i^\beta) < u^{\text{a}}_{\theta_\ell}(\pi(\beta)) = u^{\text{a}}_{\theta_\ell}(\mathbf{c}^\beta, i^\beta)$. By monotonicity of $u^{\text{a}}_{\theta_\ell}(\cdot, i^\beta)$, we have $c^\beta_{i^\beta} < h_{i^\beta} = \max\left\{0, \frac{\xi \cdot \hat{u}(\theta_\ell) - p^{\text{d}}_{i^\beta}}{r^{\text{d}}_{i^\beta} - p^{\text{d}}_{i^\beta}}, \max_{\theta \in \{\theta_1, \ldots, \theta_{\ell-1}\}} \frac{u^{\text{a}}_\theta(\pi(\theta)) - r^\theta_{i^\beta}}{p^\theta_{i^\beta} - r^\theta_{i^\beta}}\right\}$. Since $c^\beta_{i^\beta} \geq 0$, we have $h_{i^\beta} > 0$, so either (i) $c^\beta_{i^\beta} < h_{i^\beta} = \frac{\xi \cdot \hat{u}(\theta_\ell) - p^{\text{d}}_{i^\beta}}{r^{\text{d}}_{i^\beta} - p^{\text{d}}_{i^\beta}}$, or (ii) $c^\beta_{i^\beta} < h_{i^\beta} = \frac{u^{\text{a}}_\theta(\pi(\theta)) - r^\theta_{i^\beta}}{p^\theta_{i^\beta} - r^\theta_{i^\beta}}$ for some $\theta \in \{\theta_1, \ldots, \theta_{\ell-1}\}$. We show that both cases lead to contradictions.

**Case (i).** $c^\beta_{i^\beta} < \frac{\xi \cdot \hat{u}(\theta_\ell) - p^{\text{d}}_{i^\beta}}{r^{\text{d}}_{i^\beta} - p^{\text{d}}_{i^\beta}}$. It follows by monotonicity of $u^{\text{d}}(\cdot, i^\beta)$, that $u^{\text{d}}(\pi(\beta)) = u^{\text{d}}(\mathbf{c}^\beta, i^\beta) < u^{\text{d}}\left(\frac{\xi \cdot \hat{u}(\theta_\ell) - p^{\text{d}}_{i^\beta}}{r^{\text{d}}_{i^\beta} - p^{\text{d}}_{i^\beta}}, i^\beta\right) = \xi \cdot \hat{u}(\theta_\ell)$; thus, $\text{EoP}(\pi) \leq \text{EoP}_{\theta_\ell}(\pi) = \frac{u^{\text{d}}(\pi(\beta))}{\hat{u}(\theta_\ell)} < \xi$, which contradicts the assumption that $\pi$ is a satisfying policy.

**Case (ii).** $c_{i\beta}^{\beta} < \frac{u_{\theta}^{a}(\pi(\theta)) - r_{i\beta}^{\theta}}{p_{i\beta}^{\theta} - r_{i\beta}^{\theta}}$ for some $\theta \in \{\theta_1, \ldots, \theta_{\ell-1}\}$. It follows by monotonicity (decreasing) of $u^{a}(\cdot, i^{\beta})$, that $u_{\theta}^{a}(\pi(\beta)) = u_{\theta}^{a}(\mathbf{c}^{\beta}, i^{\beta}) > u_{\theta}^{a}\left(\frac{u_{\theta}^{a}(\pi(\theta)) - r_{i\beta}^{\theta}}{p_{i\beta}^{\theta} - r_{i\beta}^{\theta}}, i^{\beta}\right) = u_{\theta}^{a}(\pi(\theta))$, so a type-$\theta$ attacker would be strictly better-off reporting type $\beta$ in response to $\pi$, contradicting the assumption that $\pi$ is $(\ell - 1)$-compatible. $\qquad \square$

***Claim 2.*** $\quad h_t \leq \hat{c}_t^{\theta_\ell}$; and hence, $z_t = \min\{\hat{c}_t^{\theta_\ell}, h_t\} = h_t$.

*Proof of Claim 2.* By definition, $t \in \mathrm{BR}_{\theta_\ell}(\mathbf{h})$, so

$$u_{\theta_\ell}^{a}(\mathbf{h}, t) = \max_{i \in T} u_{\theta_\ell}^{a}(\mathbf{h}, i) \geq u_{\theta_\ell}^{a}(\mathbf{h}, i^{\beta}). \tag{16}$$

Since $\beta$ is the optimal report of a type-$\theta_\ell$ attacker, we have $u_{\theta_\ell}^{a}(\pi(\beta)) \geq u_{\theta_\ell}^{a}(\pi(\theta_\ell))$; further, by Lemma 16, $u_{\theta_\ell}^{a}(\pi(\theta_\ell)) \geq u_{\theta_\ell}^{a}(\hat{\mathbf{c}}^{\theta_\ell}, \hat{i}^{\theta_\ell})$; thus,

$$u_{\theta_\ell}^{a}(\pi(\beta)) \geq u_{\theta_\ell}^{a}(\pi(\theta_\ell)) \geq u_{\theta_\ell}^{a}(\hat{\mathbf{c}}^{\theta_\ell}, \hat{i}^{\theta_\ell}). \tag{17}$$

Combining (16), Claim 1, and (17) gives:

$$u_{\theta_\ell}^{a}(\mathbf{h}, t) \geq u_{\theta_\ell}^{a}(\mathbf{h}, i^{\beta}) \geq u_{\theta_\ell}^{a}(\pi(\beta)) \geq u_{\theta_\ell}^{a}(\hat{\mathbf{c}}^{\theta_\ell}, \hat{i}^{\theta_\ell}). \tag{18}$$

It follows that $u_{\theta_\ell}^{a}(\mathbf{h}, t) \geq u_{\theta_\ell}^{a}(\hat{\mathbf{c}}^{\theta_\ell}, \hat{i}) = \max_{i \in T} u_{\theta_\ell}^{a}(\hat{\mathbf{c}}^{\theta_\ell}, i) \geq u_{\theta_\ell}^{a}(\hat{\mathbf{c}}^{\theta_\ell}, t)$. By monotonicity of $u_{\theta_\ell}^{a}(\cdot, t)$, we have $h_t \leq \hat{c}_t^{\theta_\ell}$. $\qquad \square$

Next, we show the following parts to complete this proof: (i) $(\mathbf{z}, t)$ is indeed feasible as an outcome prescribed for report $\theta_\ell$, i.e., $\mathbf{z} \in \mathcal{C}$ and $t \in \mathrm{BR}_{\theta_\ell}(\mathbf{z})$; (ii) $\tilde{\pi}$ is $\ell$-compatible; (iii) $\mathrm{EoP}(\tilde{\pi}) \geq \xi$.

**Part (i).** Since $(\hat{\mathbf{c}}^{\theta_\ell}, i^{\theta_\ell})$ is an SSE, by definition, $\hat{\mathbf{c}}^{\theta_\ell} \in \mathcal{C}$ and $\sum_{i \in T} \hat{c}_i^{\theta_\ell} \leq m$. Since $z_i = \min\{\hat{c}_i^{\theta_\ell}, h_i\} \leq \hat{c}_i^{\theta_\ell}$ for all $i \in T$, we have $0 \leq z_i \leq 1$ and $\sum_{i \in T} z_i \leq \sum_{i \in T} \hat{c}_i^{\theta_\ell} \leq m$. Thus, $\mathbf{z} \in \mathcal{C}$.

To see that $t \in \mathrm{BR}_{\theta_\ell}(\mathbf{z})$, suppose towards a contradiction that it does not hold. Thus, $u_{\theta_\ell}^{a}(\mathbf{z}, i^{*}) > u_{\theta_\ell}^{a}(\mathbf{z}, t)$ for some $i^{*} \in T$. By Claim 2, $z_t = h_t$, so we have

$$u_{\theta_\ell}^{a}(\mathbf{z}, i^{*}) > u_{\theta_\ell}^{a}(\mathbf{z}, t) = u_{\theta_\ell}^{a}(\mathbf{h}, t) = \max_{i \in T} u_{\theta_\ell}^{a}(\mathbf{h}, i) \geq u_{\theta_\ell}^{a}(\mathbf{h}, i^{*}),$$

which implies that $z_{i^{*}} < h_{i^{*}}$ by monotonicity of $u_{\theta_\ell}^{a}(\cdot, i^{*})$. Hence, $z_{i^{*}} = \min\{\hat{c}_{i^{*}}^{\theta_\ell}, h_{i^{*}}\} = \hat{c}_{i^{*}}^{\theta_\ell}$, and

$$u_{\theta_\ell}^{a}(\hat{\mathbf{c}}^{\theta_\ell}, \hat{i}) = \max_{i \in T} u_{\theta_\ell}^{a}(\hat{\mathbf{c}}^{\theta_\ell}, i) \geq u_{\theta_\ell}^{a}(\hat{\mathbf{c}}^{\theta_\ell}, i^{*}) = u_{\theta_\ell}^{a}(\mathbf{z}, i^{*}) > u_{\theta_\ell}^{a}(\mathbf{z}, t).$$

This leads to the following contradiction:

$$u_{\theta_\ell}^{a}(\mathbf{z}, t) = u_{\theta_\ell}^{a}(\mathbf{h}, t) \geq u_{\theta_\ell}^{a}(\hat{\mathbf{c}}^{\theta_\ell}, \hat{i}) > u_{\theta_\ell}^{a}(\mathbf{z}, t),$$

where the first two (in)equalities follow by Claim 2 and 18, respectively.

**Part (ii).** We show that it is optimal for every type-$\theta$ attacker, $\theta \in \{\theta_1, \ldots, \theta_\ell\}$, to report truthfully in response to $\tilde{\pi}$.

First we consider the case for a type-$\theta_\ell$ attacker and show that $u_{\theta_\ell}^{a}(\tilde{\pi}(\theta_\ell)) \geq u_{\theta_\ell}^{a}(\tilde{\pi}(\beta'))$ for all $\beta' \in \Theta \setminus \{\theta_\ell\}$. Observe the following:

$$u_{\theta_\ell}^{a}(\mathbf{z}, t) = u_{\theta_\ell}^{a}(\mathbf{h}, t) \geq u_{\theta_\ell}^{a}(\mathbf{h}, i^{\beta}) \geq u_{\theta_\ell}^{a}(\pi(\beta)) \geq u_{\theta_\ell}^{a}(\pi(\beta')),$$

where the first three (in)equalities follow by Claim 2, (16), and Claim 1, respectively; and the last is due to the assumption that $\beta$ is the optimal reporting strategy of a type-$\theta_\ell$ attacker. By definition, $\tilde{\pi}(\theta_\ell) = (\mathbf{z}, t)$ and $\tilde{\pi}(\beta') = \pi(\beta')$ for all $\beta' \in \Theta \setminus \{\theta_\ell\}$, so $u_{\theta_\ell}^{a}(\tilde{\pi}(\theta_\ell)) = u_{\theta_\ell}^{a}(\mathbf{z}, t)$ and $u_{\theta_\ell}^{a}(\tilde{\pi}(\beta')) = u_{\theta_\ell}^{a}(\pi(\beta'))$. It follows that

$$u_{\theta_\ell}^{a}(\tilde{\pi}(\theta_\ell)) = u_{\theta_\ell}^{a}(\mathbf{z}, t) \geq u_{\theta_\ell}^{a}(\pi(\beta')) = u_{\theta_\ell}^{a}(\tilde{\pi}(\beta')),$$

which is the desired result.

Next, consider the case for each type $\theta \in \{\theta_1, \ldots, \theta_{\ell-1}\}$. By Claim 2, we have $z_t = h_t \geq \max_{\theta \in \{\theta_1, \ldots, \theta_{\ell-1}\}} \frac{u_\theta^{\mathrm{a}}(\pi(\theta)) - r_t^\theta}{p_t^\theta - r_t^\theta}$. Thus, we have

$$u_\theta^{\mathrm{a}}(\tilde{\pi}(\theta)) = u_\theta^{\mathrm{a}}(\pi(\theta)) = u_\theta^{\mathrm{a}}\left(\frac{u_\theta^{\mathrm{a}}(\pi(\theta)) - r_t^\theta}{p_t^\theta - r_t^\theta}, t\right) \geq u_\theta^{\mathrm{a}}(\mathbf{z}, t) = u_\theta^{\mathrm{a}}(\tilde{\pi}(\theta_\ell)).$$

Since $\pi$ is $(\ell-1)$-compatible, $u_\theta^{\mathrm{a}}(\pi(\theta)) \geq u_\theta^{\mathrm{a}}(\pi(\beta'))$ for all $\beta' \in \Theta$, so for those $\beta' \neq \theta_\ell$ we have

$$u_\theta^{\mathrm{a}}(\tilde{\pi}(\theta)) = u_\theta^{\mathrm{a}}(\pi(\theta)) \geq u_\theta^{\mathrm{a}}(\pi(\beta')) = u_\theta^{\mathrm{a}}(\tilde{\pi}(\beta')).$$

Therefore, $u_\theta^{\mathrm{a}}(\tilde{\pi}(\theta)) \geq u_\theta^{\mathrm{a}}(\tilde{\pi}(\beta'))$ holds for all $\beta' \in \Theta$; it is optimal for a type-$\theta$ attacker to report truthfully.

**Part (iii).** We show that $\mathrm{EoP}_\theta(\tilde{\pi}) \geq \xi$ for every type $\theta \in \Theta$, which will imply $\mathrm{EoP}(\tilde{\pi}) = \min_{\theta \in \Theta} \mathrm{EoP}_\theta \geq \xi$ and complete the proof.

Since we have shown that $\tilde{\pi}$ is $\ell$-compatible, truthful report is incentivized for every type in $\{\theta_1, \ldots, \theta_\ell\}$. Thus, for every $\theta \in \{\theta_1, \ldots, \theta_\ell\}$, we have $\mathrm{EoP}_\theta(\tilde{\pi}) \geq \frac{u^{\mathrm{d}}(\tilde{\pi}(\theta))}{\hat{u}(\theta)}$. (The reason we have an inequality here is due to the optimistic tie-breaking assumption in Definition 6.) For type $\theta_\ell$, since $z_t = h_t \geq \frac{\xi \cdot \hat{u}(\theta_\ell) - p_t^{\mathrm{d}}}{r_t^{\mathrm{d}} - p_t^{\mathrm{d}}}$ by Claim 2, we have

$$\mathrm{EoP}_{\theta_\ell}(\tilde{\pi}) \geq \frac{u^{\mathrm{d}}(\tilde{\pi}(\theta_\ell))}{\hat{u}(\theta_\ell)} = \frac{u^{\mathrm{d}}(\tilde{\pi}(\mathbf{z}, t))}{\hat{u}(\theta_\ell)} \geq \frac{u^{\mathrm{d}}\left(\frac{\xi \cdot \hat{u}(\theta_\ell) - p_t^{\mathrm{d}}}{r_t^{\mathrm{d}} - p_t^{\mathrm{d}}}, t\right)}{\hat{u}(\theta_\ell)} = \frac{\xi \cdot \hat{u}(\theta_\ell)}{\hat{u}(\theta_\ell)} = \xi.$$

For types $\theta \in \{\theta_1, \ldots, \theta_{\ell-1}\}$, we have

$$\mathrm{EoP}_\theta(\tilde{\pi}) \geq \frac{u^{\mathrm{d}}(\tilde{\pi}(\theta))}{\hat{u}(\theta)} = \frac{u^{\mathrm{d}}(\pi(\theta))}{\hat{u}(\theta)} = \mathrm{EoP}_\theta(\pi) \geq \mathrm{EoP}(\pi) \geq \xi.$$

For the other types $\theta' \in \{\theta_{\ell+1}, \ldots, \theta_\lambda\}$, if their optimal report remains the same as under $\pi$, for the same argument above, $\mathrm{EoP}_{\theta'}(\tilde{\pi}) = \mathrm{EoP}_{\theta'}(\pi) \geq \mathrm{EoP}(\pi) \geq \xi$. Otherwise, since $\tilde{\pi}$ and $\pi$ differs only in the outcomes prescribed for type $\theta_\ell$, if the attacker's optimal reporting strategy changes under $\tilde{\pi}$, it will only change to $\theta_\ell$, in which case we have

$$\mathrm{EoP}_{\theta'}(\tilde{\pi}) = \frac{u^{\mathrm{d}}(\tilde{\pi}(\theta_\ell))}{\hat{u}(\theta')} \geq \frac{u^{\mathrm{d}}(\tilde{\pi}(\theta_\ell))}{\hat{u}(\theta_\ell)} = \mathrm{EoP}_{\theta_\ell}(\tilde{\pi}) \geq \xi,$$

where the first inequality holds because Algorithm 1 orders attacker types in a way such that $\hat{u}(\theta') \leq \hat{u}(\theta_\ell)$ for all $\theta' \in \{\theta_{\ell+1}, \ldots, \theta_\lambda\}$. $\qquad\square$

## D   Complexity of Computing Optimal Policy in *General* Stackelberg Games

We show the complexity of computing the optimal leader policy in *general* Stackelberg games where payoff parameters of each player (or player type) are given by a matrix, with no restriction on the values. We let $u^{\mathrm{L}} \in \mathbb{R}^{m \times n}$ denote the leader's payoff matrix, and $u_\theta^{\mathrm{F}} \in \mathbb{R}^{m \times n}$ denote a type-$\theta$ follower's payoff matrix for each follower type $\theta \in \Theta$, where $m$ and $n$ denote the numbers of the leader's and the follower's actions (i.e., pure strategies), respectively. The entries $u^{\mathrm{d}}(i, j)$ and $u_\theta^{\mathrm{a}}(i, j)$ are, respectively, the utilities of the leader and a type-$\theta$ follower, when the leader plays her $i$-th action and the follower plays his $j$-th action. In an SSE, the leader plays a mixed strategy $\mathbf{x} \in \Delta_m$ and the follower best responds to $\mathbf{x}$ with a pure strategy $j$, yielding leader utility $u^{\mathrm{L}}(\mathbf{x}, j) = \sum_{i=1}^m x_i \cdot u^{\mathrm{L}}(i, j)$ and follower utility $u^{\mathrm{F}}(\mathbf{x}, j) = \sum_{i=1}^m x_i \cdot u^{\mathrm{F}}(i, j)$. All other definitions and notation are the same as in Section 2. In contrast to the tractability of computing the optimal defender policy in an SSG, the problem is hard in general Stackelberg games.

**Theorem 17.** *It is NP-complete to decide whether there exists a leader policy $\pi$ with $\mathrm{EoP}(\pi) \geq \xi$.*

*Proof.* The NP membership of the problem is straightforward as for any given policy $\pi$, we can efficiently verify whether $\mathrm{EoP}(\pi) \geq \xi$. For the NP-hardness, we show a reduction from the VERTEX

|  | $1^F$ | $2^F$ | $3^F$ |
|---|---|---|---|
| $\forall i$ | 1 | 0.5 | |

leader

| | $1^F$ | $2^F$ | $3^F$ |
|---|---|---|---|
| $\forall i \in \{a_{v_1}, a_{v_2}\}$ | 0.9 | 0.9 | 1 |
| $a_0$ | 0.4 | | 0.4 |
| *otherwise* | | 0.9 | 1 |

flw. type $\theta_e$
$(e = \{v_1, v_2\} \in E)$

| | $1^F$ | $2^F$ | $3^F$ |
|---|---|---|---|
| $\forall i$ | 1 | | |

flw. type $\theta_\ell$
$(\ell = 1, \dots, k)$

| | $1^F$ | $2^F$ | $3^F$ |
|---|---|---|---|
| $a_0$ | | | 1 |
| $\forall i \neq a_0$ | | 1 | 1 |

flw. type $\theta_0$

Figure 2: Payoff parameters (blank entries are all 0).

COVER problem, which is well-known to be NP-complete. A *vertex cover* $V'$ of an undirected graph $G = (V, E)$ is a subset of $V$ such that $v_1 \in V'$ or $v_2 \in V'$ for every edge $\{v_1, v_2\} \in E$. An instance of the VERTEX COVER problem is given by a graph $G = (V, E)$ and an integer $k \leq |V|$. It is a yes-instance if there exists a vertex cover of $G$ of size at most $k$.

For a VERTEX COVER instance, we construct the following game and show that the VERTEX COVER instance is a yes-instance if and only if there exists a leader policy $\pi$, $\mathrm{EoP}(\pi) \geq 1$. In the game, the leader has $|V| + 1$ actions $\{a_v : v \in V\} \cup \{a_0\}$. The follower has three actions $\{1^F, 2^F, 3^F\}$. The set of possible follower types is $\Theta = \{\theta_1, \dots, \theta_k\} \cup \{\theta_e : e \in E\} \cup \{\theta_0\}$. The payoffs are given in Figure 2.

We first make several observations about the SSEs of this game (in the truthful situation). Below we let $\hat{u}^L(\theta)$ denote the leader's utility in an SSE when she plays against a type-$\theta$ follower, i.e., $\hat{u}^L(\theta) = \max_{\mathbf{x} \in \Delta_m, j \in \mathrm{BR}(\mathbf{x})} u^L(\mathbf{x}, j)$.

- The leader's utility only depends on the follower's action, with $3^F$ being the most detrimental action the leader would anyhow avoid the follower to choose, followed by $2^F$, and $1^F$ is the most preferred follower action.

- The *only* SSE strategy of the leader when she plays against a type-$\theta_e$ follower, $e \in E$, is the pure strategy $a_0$. When $a_0$ is played, the follower finds his best responses to be $\mathrm{BR}_{\theta_s}(a_0) = \{3^F, 1^F\}$ and breaks the tie in favor of the leader, playing $1^F$; the leader obtains $u^L(a_0, 1^F) = 1$, which is obviously the highest possible utility she can obtain; hence, $(a_0, 1^F)$ forms an SSE and $\hat{u}^L(\theta_e) = 1$. To see that this is the only SSE, observe that if the leader plays any other pure strategy with some probability, the follower would strictly prefer $3^F$ to $1^F$ and would not respond by playing $1^F$, in which case the leader cannot get utility 1.

- Every leader strategy is an SSE strategy when the follower she plays against has type $\theta_\ell$, $\ell = 1, \dots, k$, because the follower will always respond by playing $1^F$ irrespective of the strategy the leader plays, which always gives the leader utility 1. We have $\hat{u}^L(\theta_\ell) = 1$.

- Every mixed leader strategy over pure strategies $i \neq a_0$ is an SSE strategy of the leader when the follower she plays against has type $\theta_0$. When such a strategy is played, the follower finds his best response set to be $\{3^F, 2^F\}$ and breaks the tie in favor of the leader, playing $2^F$; this is the best the leader can hope for because a type-$\theta_0$ follower will never play $1^F$ as it is strictly dominated by $3^F$. We have $\hat{u}^L(\theta_0) = 0.5$.

Suppose that there exists a vertex cover $V' = \{v'_1, \ldots, v'_k\} \subseteq V$ of size $k$. The following leader policy $\pi$ achieves EoP 1.

$$\pi(\theta) = \begin{cases} (a_0, \ 1^{\mathrm{F}}), & \text{for each } \theta \in \{\theta_e : e \in E\}; \\ (a_{v'_\ell}, \ 1^{\mathrm{F}}), & \text{for each } \theta = \theta_\ell \in \{\theta_1, \ldots, \theta_k\}; \\ (a_{v'_1}, \ 2^{\mathrm{F}}), & \text{for } \theta = \theta_0. \end{cases}$$

Clearly, all the outcomes prescribed are feasible, so $\pi$ is a feasible policy. Further, it can be verified that when the leader commits to $\pi$, the optimal reporting strategy of the follower is the following.

- For every type-$\theta_e$ follower, $e \in E$, it is optimal to report a type $\theta_\ell$ with $v'_\ell \in V'$ bing an end point of $e$. Such a $\theta_\ell$ always exists given that $V'$ is a vertex cover. The leader obtains utility 1 when $\theta_\ell$ is reported, so $\mathrm{EoP}_{\theta_e}(\pi) \geq \frac{1}{\hat{u}(\theta_e)} = 1$.

- For every type-$\theta_\ell$ follower, $\ell \in \{1, \ldots, k\}$, it is optimal to report truthfully. The leader obtains utility 1 when $\theta - \ell$ is reported, so $\mathrm{EoP}_{\theta_\ell}(\pi) \geq \frac{1}{\hat{u}(\theta_\ell)} = 1$.

- For a type-$\theta_0$ follower, it is optimal to report truthfully. The leader obtains utility 0.5 when $\theta_0$ is reported, so $\mathrm{EoP}_{\theta_0}(\pi) \geq \frac{0.5}{\hat{u}(\theta_0)} = 1$.

Therefore, $\mathrm{EoP}(\pi) = \min_{\theta \in \Theta} \mathrm{EoP}_\theta(\pi) = 1$. (We have $\mathrm{EoP}(\pi) \leq 1$ by Proposition 7; in fact, $\mathrm{EoP}_\theta(\pi)$ is also upper-bounded by 1 for types $\theta_e$ and $\theta_\ell$ above.)

Conversely, suppose that there exists a policy $\pi$ with $\mathrm{EoP}(\pi) = 1$. We show that there exists a vertex cover of $G$ of size at most $k$. For $\mathrm{EoP}(\pi) = 1$, we need $\mathrm{EoP}_\theta(\pi) \geq 1$ for all $\theta \in \Theta$. Thus, the actual utility the leader obtains must be: at least 1 on each $\theta_e$ and $\theta_\ell$ ($\ell \neq 0$), and at least 0.5 on $\theta_0$.

Now consider the reporting strategy of a type-$\theta_0$ follower in response to $\pi$; let $\beta \in \Theta$ be the optimal reporting strategy of a type-$\theta_0$ follower, and let $\pi(\beta) = (\mathbf{x}^\beta, j^\beta)$. For the leader to obtain actual utility at least 0.5 on type $\theta_0$, we need $j^\beta \in \{1^{\mathrm{F}}, 2^{\mathrm{F}}\}$. Observe that a type-$\theta_0$ follower gets utility 0 if $j^\beta = 1^{\mathrm{F}}$, in which case he would be better-off reporting truthfully to avoid being induced to "best" respond by playing $1^{\mathrm{F}}$. Thus, the only possibility is $j^\beta = 2^{\mathrm{F}}$. Now that $j^\beta = 2^{\mathrm{F}}$, it must also be that $x^\beta_{a_0} = 0$ since otherwise the follower obtains less than 1 by reporting $\beta$ and would, again, be better-off reporting truthfully (in which case he is guaranteed utility 1 by the best response $3^{\mathrm{F}}$).

Given this, a type-$\theta_e$ follower, $e = \{v_1, v_2\} \in E$, is able to obtain utility 0.9 by reporting $\beta$. Let $\gamma$ be the type a type-$\theta_e$ follower is incentivized to report, and $\pi(\gamma) = (\mathbf{x}^\gamma, j^\gamma)$; we therefore have $u^{\mathrm{F}}_{\theta_e}(\mathbf{x}^\gamma, j^\gamma) \geq 0.9$. For the leader to obtain utility at least 1 on type $\theta_e$, we need $j^\gamma = 1^{\mathrm{F}}$, in which case $u^{\mathrm{F}}_{\theta_e}(\mathbf{x}^\gamma, j^\gamma) \geq 0.9$ only if $x^\gamma_{a_{v_1}} + x^\gamma_{a_{v_2}} = 1$ (i.e., only the first row of the payoff matrix of $\theta_e$ is chosen), so we have $x^\gamma_{a_0} = 0$. Therefore, $\gamma \notin \{\theta_{e'} : e' \in E\}$, because this would lead to $\mathrm{BR}_\gamma(\mathbf{x}^\gamma) = \{3^{\mathrm{F}}\} \not\ni 1^{\mathrm{F}}$ given that $x^\gamma_{a_0} = 0$. For the same reason, we also have $\gamma \neq \theta_0$, so the remaining possibility is that $\gamma = \theta_\ell$ for some $\ell \in \{1, \ldots, k\}$.

Let $V^\gamma = \{v \in V : x^\gamma_{a_v} \geq 0\}$, and let $v'_\gamma$ be the first vertex in $V^\gamma$ in lexicographical order. Since $x^\gamma_{a_{v_1}} + x^\gamma_{a_{v_2}} = 1$, we have $v'_\gamma \in e$, and moreover, $\{v'_{\theta_\ell} : \ell = 1, \ldots, k\} \cap e \neq \varnothing$ given that $\gamma = \theta_\ell$ for some $\ell$. This holds for all $e \in E$. Thus, $V' = \{v'_{\theta_\ell} : \ell = 1, \ldots, k\}$ forms a vertex cover of $G$, and $|V'| \leq k$. $\qquad \square$

## E    Additional Experiment Results

**EoP Comparison**    Figures 3 and 4 show additional results of the EoP comparison. In both figures, (a)–(c) show the variance of EoP with respect to $\rho$, with type sets of different scales ($\lambda = 10, 100$, and 1000, respectively); (d)–(f) show the variance of the EoP with respect to the scale of the game, under different target-resource ratios ($\frac{n}{m} = 10, 5$, and 2, respectively). In Figure 3, attacker types are generated with the covariance model, while in Figure 4, the zero attacker type is always included in $\Theta$ in addition to types generated by the covariance model.

Figure 3: Comparison of the EoP. Other parameters are set to $n = 50$, $m = 10$ in (a)–(c); and $\rho = 0.5$, $\lambda = 100$ in (d)–(f).

Figure 4: Comparison of the EoP when the zero-sum attacker type is always included in $\Theta$ in addition to types generated by the covariance model. Other parameters are specified in the same way as in Figure 3.

**Algorithm Runtime** Figure 5 shows results of the runtime test of our algorithms. All results are obtained on a platform with a 2.60 GHz CPU and a 8.0 GB memory. The time for computing SSEs is excluded in the results as this is handled by an existing algorithm, the performance of which is not our focus. Both our algorithms for computing the optimal policy and the QR policy exhibit good scalability, capable of solving problems of 5000 attacker types and 500 targets in a reasonable amount of time. The computation of QR policy is extremely efficient thanks to its simplicity.

(a) Optimal policy

| | $n : 100$ | 200 | 300 | 400 | 500 |
|---|---|---|---|---|---|
| $\lambda : 1000$ | 3.54 | 7.45 | 10.65 | 14.26 | 17.96 |
| 2000 | 14.84 | 30.81 | 46.84 | 75.99 | 78.75 |
| 3000 | 32.42 | 63.50 | 96.92 | 126.96 | 161.37 |
| 4000 | 57.47 | 115.05 | 171.00 | 228.20 | 288.19 |
| 5000 | 88.84 | 184.77 | 273.51 | 365.91 | 480.54 |

(b) QR policy

| | $n : 100$ | 200 | 300 | 400 | 500 |
|---|---|---|---|---|---|
| $\lambda : 1000$ | 0.00 | 0.02 | 0.02 | 0.03 | 0.03 |
| 2000 | 0.01 | 0.03 | 0.04 | 0.06 | 0.10 |
| 3000 | 0.02 | 0.05 | 0.07 | 0.10 | 0.12 |
| 4000 | 0.03 | 0.06 | 0.09 | 0.12 | 0.16 |
| 5000 | 0.04 | 0.08 | 0.10 | 0.16 | 0.20 |

Figure 5: Algorithm runtime (seconds).

## Footnotes

[4] A solution $x$ satisfies stationarity if $\nabla f(x) = \lambda_1 \cdot \nabla g_1(x) + \cdots + \lambda_\ell \cdot \nabla g_\ell(x)$, where $f$ is the objective function (minimization), each $g_i$ corresponds to an inequality constraint (in the form $g_i(x) \leq 0$), and each $\lambda_i$ is a KKT multiplier.

[5] A solution $x$ satisfies complementary slackness if $\lambda_i \cdot g_i(x) = 0$ for each KKT multiplier $\lambda_i$ and their corresponding inequality constraint function $g_i$.