[Reviews · NeurIPS 2019]

Reviewer 1



The paper is written well; analysis is clear and results are reasonable. Nevertheless, the model setup is rather simple, lacking of real problem illustration. Also, the learning process for the defender is missing.

Reviewer 2



Clarity: The paper is a pleasure to read. The motivation is clear and the problem is well situated with respect to other work in the literature. The paper is self contained and in general, flows well. Due to space constraints, some of the content such as simulation results were deferred to the appendix. Quality: The approach and derivations shown in the main paper appear to be sound and correct. That said, I did not scrutinize the derivations given in the appendix. Clarifications needed: (1) It is claimed in the introduction that the problem of finding the optimal EoP can be done in polynomial time by performing binary search on \xi. Does `polynomial’ here include being polynomial with respect to the true optimal (1/\xi)? If not, could one possible construct a particular set of payoffs which has arbitrarily small \xi, that is, when binary search is applied, could take arbitrarily many iterations. (2) The setting in section 6 is unclear to me. The motivation behind the QR policy is allowing for a continuous set of attacker types. On the other hand, the model being considered is strictly more general than in previous section (allowing for randomized commitments), and is reported to obtain better utility than the “perfectly rational” agent. In this setting, is the attacker aware of the bounded rationality behind the defender? If so, is the optimal report of the attacker (assuming the attacker type is the space of all possible attacker payoffs) still one that makes the “fake” game zero-sum? Furthermore, is there any particular reason why the authors selected the QR policy (other than convenience?). There are other models for bounded rationality that result in randomized outcomes Originality: The problem setting and the contributions listed earlier are novel to the best of my knowledge. Significance: The paper shows the counterintuitive that a manipulative attacker is incentivized to report payoffs which make the game zero-sum. The EoP metric is sensible and could be of interest to the game theory or multi-agent communities, in particular, those who work on deception. The setting of a manipulative attacker featured here is (to the best of my knowledge) fairly novel. Other comments: (3) There appears to be a typo in Example 1: the payoffs should be r_A = 3 and r_B = 1; in the text, these values are swapped. (4) It is known that stackelberg games may be reformulated (albeit in a clunky fashion) as an extensive form game (see the manuscript “On Stackelberg Mixed Strategies” by Vincent Conitzer). It appears (correct me if I am mistaken) that the problem setting considered in this paper could be given the same treatment. This conversion to extensive form games could also be performed for the setting with randomized defender policies. Could the authors provide some insight here? =============================== Post rebuttal: The authors have addressed my concerns adequately. That said, I maintain that section 6 is poorly written and should be significantly improved, alongside the other recommended changes. As such, I am keeping my score unchanged.

Reviewer 3



Originality ------------ The paper appears to be fairly original with respect to the SSG literature. However, my primary concern is that it does not reference the mechanism design literature -- and appears to be highly related to that line of research. In particular, one way to describe the contributions of this work are that they extend the analysis of SSGs to the case of multiple unknown types and account for the incentive for the attacker to lie about his type. For example, a bit of time on Google scholar pointed me to, e.g., "A Dynamic Bayesian Security Game Framework for Strategic Defense Mechanism Design" and "Design of Incentive Compatible Mechanisms for Stackelberg Problems." Each of these deal with truthful reporting and Stackelberg games (although the setting is somewhat different from the setting considered here). In order to fully evaluate the originality of this work, the authors should update the paper to include a brief review of this line of work and exlicity compare/contrast it to their contributions here. Quality -------- The theoretical results are well motivated, and the proofs are sound. The overall quality of the paper is quite good, with the fairly stark exception of section 6. The authors introduce quantal response equilibria as a way to handle infinite attacker types and refer to a set of simulations on random games in order to argue for its performance. I do not see the value in this section, other than pointing our that this strategy can be computed for infinite type sets. The experiments are referred to with so little detail that the reader simply can not get the full picture without referencing the supplementary materials. The authors then refer to the surprising result that the QR policy achieves better performance than the optimal response. This is somewhat surprising (although possible due to the entropy maximizing nature of the softmax distribution used), but the main issue is that it does not appear to be supported by the figures in the supplementary material. From my look at figures B.2 and B.3, there is only a small region of one graph where the QR policy outperforms the optimal policy. Overall, this section would be improved by removing the experimental results as they currently stand --- evaluated as a stand alone work, they are too vague to make a real argument and they appear to mischaracterize the data in the supplementary material. To me, the paper would be better served by either removing this section (to free up space to improve the theoretical results), or figuring out a way to include a more complete representation of the experiments. Alternatively, this section could be reframed to discuss the relative performance of the optimal strategy from section 5 and the SSE solution, which lines up better with the contributions in the introduction. Clarity ------- Overall, the paper is reasonably well-written. There are several places where the paper's readability can be improved, I've highlighted the major areas for improvement below and have some smaller comments at the bottom of the review. 164: Is this program for the optimal sequence of responses? or just for the next report? Looking back, I think I can see where I got confused --- this is representing the full interaction and the disclaimers above allow for this simplified representation. Perhaps it would help to link the form of this program to the assumptions above and explain how they help to simplify the representation. In many ways, this formulation is a key contribution of the paper and so it is important to clarify how it is an improvement over the naive formulation of learning. 176: I think it could be good to redo the reduced version of the proof for the paper, focusing more on conveying the intuition and steps in the proof at a more abstract level. Right now, it is quite notation dense and does not convey much information to the reader beyond the need to look in the supplementary material. To some extent, the proofs in the supplementary material could use some clarification as well to make them accessible to a broader audience. For example, consider providing some intuition behind the maximin strategy profile used or explaining the intuition behind why the maximin performance can't be improved in the final proof step. 209: This is a good example, but I think it can be improved by 1) clarifying here that this is bad for the defender because they would really prefer to attack A. I also wonder about this strategy, because it seems to rely on an assumption the A is more likely to be more valuable than B? It seems like this type of approach will depend on the structure of the set of attacker profiles $\Theta$ (this is mentioned in 232, but would be good to bring earlier in the writing IMO) 239: this explanation is hard to follow, it seems like the assumption about $\Theta$ being the true types is out of place. Consider being more specific about the definition earlier on: e.g., EoP is the ratio of the defenders utility under the policy to the utility the would get if they knew the true type. It is less than 1 because.... A higher EoP means that the defender is closer to matching the utility achieved when the attacker's type is known. Significance -------------- I believe the paper here is significant in that it identifies and clearly illustrates a clear gap in some of the existing literature on Stackelberg games. The authors effectively argue that this gap is significant. On that basis, I believe that the paper is worthy of acceptance. However, I believe that the current presentation does not do an adequate job of comparing the work to existing literature and it makes it hard to judge the significance of the additional contributions in the work. In particular, the authors can do a better job of arguing why their proposed EoP measure is a good measurement of performance in this setting as the utility of their algorithm and the implications of their experimental results both hinge on this. Other Suggestions --------------------- 142: "resulting in his utility to increase" 148: clarify the significance of these results for a more general audience: why is playing the maximin strategy bad? 157: A little hard to follow here. Maybe, consider using alternative word or phrase than "conforms," which is doing a lot of work here. 232: "we face the following challenges" --- the list of challenges is separated across multiple paragraphs so this list increases cognitive load on the reader and hurts clarity. I would get rid of the rhetorical question; list the two challenges here; and then describe them in more detail in subsequent paragraphs. 238: "termed" awk, rephrase 288: "will act truthfully" --- I think this should be 'untruthfully'? 246: move this proposition to be closer to the text it is referenced by 252: maintain consistent capitalization for EoP_\theta

Reviewer 4



The paper is well written and the mathematics looks correct. The technical presentation of the paper is concise, and concrete examples are used to illustrate the fundamental ideas. All theoretical results are supplemented with complete proofs. Also such research could shed some light on poissoning attacks in adversarial machine learning, a current hot subject. Limitations: - The empirical results should be moved from the supplemental, to the main paper. As the paper stands, its presentation is incomplete. Whole sections of a paper shouldn't be moved to supplemental in order to make space. Page limits exist for a reason and should be respected. - One thing that bugs me is that the authors assume there is such a thing as a "learning phase" and "testing phase". I doubt such a division of concerns is reasonable in real-life. Also, assuming that the space of possible attack types for the attacker / follower is public known is rather a very restricted assumption. In "real-life", a defender would eventually learn the attacker's type modulo statistical noice associated to the fact that only a finite number of interactions are available. Thus a defender could replace empirically learned follower / attacker type with an uncertainty set around it (as is done in "distributionally robust optimization" literature), and play against an imaginary attacker with worst-case type in this uncertainty set. This would be optimal and data-driven. No ? - The new proposed goodness measure EoP, seems somewhat adhoc. What interesting conceptual / theoretical properties does EoP have ? A theoretical comparison to more classical measures would be worthwhile. Typos: - line 57: improvement ==> bring improvement ? - line 64: please give concrete examples of such "many real-world" situations

[Author Response · NeurIPS 2019]

We thank all reviewers for their very helpful comments. We'll fix all typos and minor issues, and incorporate the
suggested changes. Because of space constraints, we only focus on answering the reviewers' major questions below.

**Reviewer 1.** Our model is a standard Stackelberg security game (SSG), which forms the basis of many real-world
applications in security domains. For example, in the Los Angeles International Airport, the SSG model is adopted to
help decide allocation of canine units to terminals, and similar systems are adopted by the US Federal Air Marshal
Service to deploy armed marshals to commercial flights [Jain-An-Tambe 2011 AI Magazine]. There is a large body of
literature on SSGs with many model variants developed for specific scenarios [Tambe 2011]. We focus on the standard
(and also the most general) model as the first work on attacker manipulation; the choice of model is in line with previous
works on designing defender learning algorithms (e.g., [Blum-Haghtalab-Procaccia NIPS'14; Peng *et al.* AAAI'19]).

Our policy-based framework wraps the defender's learning algorithm as a sub-procedure, and allows *any* learning
algorithm to be used as this sub-procedure as long as the algorithm learns by observing the attacker's best responses
(e.g., algorithms in [Letchford-Conitzer-Munagala SAGT'09; Blum-Haghtalab-Procaccia NIPS'14; Haghtalab *et al.*
IJCAI'16; Peng *et al.* AAAI'19]). The actual learning process is therefore abstracted as a reporting stage in our paper.

**Reviewer 2.** Binary search finds the optimal EoP $\xi$ within any desired precision $\epsilon > 0$ in time $O(\log(\frac{1}{\epsilon}))$. It terminates
when we can locate $\xi$ in a small interval $[a, a + \epsilon]$. We believe this is a common approach for similar searching problems
on a continuous range. It would be too demanding to seek the *exact* $\xi$ even in the theoretical sense, as it's unclear
whether the optimal $\xi$ is a rational number to allow for a computationally feasible representation in the first place.

In the QR setting, the attacker is still aware of the "bounded rationality" of the defender. All assumptions are the same
as the previous sections, except that here the defender is allowed to further randomize her commitment. In particular,
when we say "perfectly/boundedly rational behavior", we refer to perfectly/boundedly rational behavior **in the truthful**
**setting** where the attacker does **not** manipulate (we'll clarify this in the paper). When the attacker does manipulate,
however, such "perfectly rational" behavior may turn out to be suboptimal and even worse than "boundedly rational"
behavior. Intuitively, this is because the "perfectly rational" behavior falls into a fixed pattern for the attacker to exploit,
whereas the "boundedly rational" behavior adds uncertainty in the fixed pattern and complicates attacker manipulation,
in which case the attacker's old trick of making the fake game zero-sum would not work anymore.

We choose the QR model because it strikes a balance between the following two unaligned aspects of playing against
attacker manipulation: 1) we want to discourage/punish attacker manipulation; 2) meanwhile we don't want the cost
of 1) to be too high for the defender. The QR model on the one hand adds uncertainty in the defender's behavior that
discourages attacker manipulation to some extent, while on the other hand it approximates the optimal strategy by
choosing better actions with higher probabilities (i.e., the *softmax* function approximates the *max* function). We say that
this is the rationality of QR (as in the **un**truthful setting) behind its bounded rationality (as in the truthful setting).

Stackelberg games are two-stage *extensive form games* (EFG). Technically, however, there's perhaps not much benefit
of approaching our model as an EFG (as mentioned in Conitzer's paper, the defender's strategy space is infinite so there
will be infinitely many nodes on the game tree), but it looks very interesting to study similar manipulation in EFGs.

**Reviewer 3.** We fully agree that our leader policy design can be viewed as a mechanism design problem and appreciate
the reviewer pointing out the connection. Our work differs from other mechanism design problems in the underlying
SSG that decides the players' utilities, where the mechanism designer plays as the defender and the attacker can lie about
their payoffs. Our contribution lies in applying mechanism design ideas to this specific setting to tackle manipulation to
defender learning algorithms. We'll add relevant discussions to make both the connection and the comparison.

For the QR model in Section 6, we refer the reviewer to our response to Reviewer 2. In the experiments, results of
the "perfectly rational" behavior are represented by the **green** curves and labeld "SSE" (since SSE corresponds to the
strategy of a perfectly rational defender *in the truthful setting*). Thus, the QR policy performs better than SSE in most
cases, and only sometimes better than the optimal policy (**red** curves). The reason it can perform better than the optimal
policy is that it allows additional randomization in the policy commitment (also see footnote 4 in the paper).

We introduce EoP because the *worst-case utility* is unable to distinguish the quality of many policies. As shown in
Proposition 5, when playing against an attacker type whose payoffs are zero-sum to the defender's payoffs, no policy
can achieve anything better than the *maximin utility* — essentially, it's a "mission impossible" to play well against such
a zero-sum attacker type. As a result, if we take the worst-case utility as the criterion, the quality of all feasible policies
would be hindered on this zero-sum attacker type, and hence be considered to be *no* better than the *SSE policy* (defined
in Line 226); there will be no room for improvement against this criterion other than letting the attacker manipulate.
This is unreasonable: *we cannot simply consider a policy to be bad just because it underperforms in some "mission*
*impossible"*. The EoP is therefore proposed to adjust our measurement by taking into consideration also *the degree*
*of difficulty* of the "missions" that is measured by the best achievable defender utility in the truthful setting (as the
denominator of $\text{eop}_\theta$, Line 252). There is no additional assumption made about the attacker's abilities when EoP is
defined. Each attacker type always aims at maximizing their absolute utility. We'll update the paper accordingly.

[Meta-Review · NeurIPS 2019]

In spite of several flaws regarding the presentation, the contribution has been judged sufficiently novel and sound to recommend acceptance.